materials science

gas separation, mixed matrix membrane, polysulfone-zeolite templated carbon, annealing, coating

**Author for correspondence:**
Nurul Widiastuti
e-mail: nurul_widiastuti@chem.its.ac.id

This article has been edited by the Royal Society of Chemistry, including the commissioning, peer review process and editorial aspects up to the point of acceptance.

# Annealing and TMOS coating on PSF/ZTC mixed matrix membrane for enhanced $CO_2/CH_4$ and $H_2/CH_4$ separation

Nurul Widiastuti[1], Irmariza Shafitri Caralin[2],
Alvin Rahmad Widyanto[1], Rika Wijiyanti[3],
Triyanda Gunawan[1], Zulhairun Abdul Karim[4],
Mikihiro Nomura[2] and Yuki Yoshida[2]

[1]Department of Chemistry, Faculty of Science and Data Analytics, Institut Teknologi Sepuluh Nopember, Sukolilo, Surabaya 60111, Indonesia
[2]Department of Applied Chemistry, Shibaura Institute of Technology, 3-7-5 Toyosu, Koto-ku, Tokyo 135-8548, Japan
[3]Medical Intelligence, Sekolah Tinggi Intelijen Negara (State Intelligence College), Sumur Batu, Babakan Madang, Bogor 16810, Indonesia
[4]Advanced Membrane Technology Research Center (AMTEC), Universiti Teknologi Malaysia, 81310 UTM Skudai, Johor Darul Ta'zim, Malaysia

NW, 0000-0002-6821-8116; TG, 0000-0003-0836-7715

Recently, natural gas (mostly methane) is frequently used as fuel, while hydrogen is a promising renewable energy source. However, each gas produced contains impurity gases. As a result, membrane separation is required. The mixed matrix membrane (MMM) is a promising membrane. The huge surface area and well-defined pore structure of zeolite templated carbon (ZTC)-based MMM allow for effective separation. However, the interfacial vacuum in MMM is difficult to avoid, contributing to poor separation performance. This research tries to improve separation performance by altering membrane surfaces. MMM PSF/ZTC was modified by annealing at 120, 150, and 190°C; coating using 0.01, 0.03, and 0.05 mol tetramethyl orthosilicate (TMOS); and a combination of both, i.e. annealing at 190°C and coating using 0.03 mol TMOS. MMM PSF/ZTC successfully significantly improved $CO_2/CH_4$ selectivity by a combination of annealing at 190°C and coating 0.03 mol TMOS from 1.37 to 5.90 (331%), and $H_2/CH_4$ selectivity by coating with 0.03 mol TMOS from 4.58 to 65.76 (1378%). The enhancement of selectivity was due to structural changes to the membrane that became denser and smoother, which SEM and AFM observed. In this study, annealing and coating treatments are the methods investigated for improving the polymer matrix and filler particle adhesion.

# 1. Introduction

Natural gas, which mostly consists of methane ($CH_4$), is a widely used fuel that can play an important role as a complementary transition fuel promoting renewable energy during the transition phases [1]. In raw natural gas, there is an impurity in the form of $CO_2$, which can cause corrosion in pipes and decrease the heating value of natural gas [2]. Other gases such as hydrogen also have potential as renewable energy, which can be used as a substitute for coal [3–5]. In its use as a fuel, hydrogen is environmentally friendly because it does not produce pollutant emissions. Among all hydrogen production technologies [6–8], steam reforming gas is one that is commonly used with $CH_4$ feedstock [9]. But in the process, not all $CH_4$ can be converted properly into $H_2$ gas, so the produced $H_2$ gas is not pure [10]. $CH_4$ impurities can cause a reduction in catalyst performance when $H_2$ is used for fuel cells [11]. Therefore, technological innovation is needed to separate the gases.

Membrane technology offers several advantages over conventional technologies in gas separation. Conventional technologies such as cryogenic distillation, evaporation, absorption, and drying have the disadvantages of requiring a large amount of energy and producing pollution [12]. The advantages of membrane technology are high energy efficiency, a continuous and straightforward operating system, relatively low cost, and environmental friendliness [13]. Compared with traditional distillation processes, the separation process using membranes requires about 90% less energy [14]. Membrane technology is promising in gas separation applications. Thus, it needs to be developed and researched.

The polymer membrane is a material that is currently widely used in large-scale industry for gas separation processes because of its good mechanical properties and flexibility [15–17]. However, the gas separation performance, which is affected by the trade-off between permeability and selectivity as shown by the Robeson curve, is one of the weaknesses of polymer membranes [18]. In Pakizeh and Hokmabadi's research [19], polysulfone membranes were used for the separation of $CO_2/CH_4$ and $H_2/CH_4$ gases, with permeability of $CO_2$, $H_2$, and $CH_4$ of 4.77, 7.49, and 0.26 Barrer, and the selectivity of $CO_2/CH_4$ and $H_2/CH_4$ of 18.35 and 28.81, respectively, but showed poor separation performance when compared with the Robeson curve [20]. On the other hand, inorganic membranes have several advantages such as high thermal and chemical stability, as well as excellent separation performance. However, several disadvantages such as high operational costs and difficult operation can be considered in their application [21]. In the research of Favvas *et al*. [22], the carbon membrane used for the separation of $CO_2/CH_4$ and $H_2/CH_4$ gases has a performance above the Robeson curve, with $CO_2$, $H_2$, and $CH_4$ permeabilities of 6.79, 36.49, and 0.37 GPU, and selectivity for $CO_2/CH_4$ and $H_2/CH_4$ of 18.35 and 98.62, respectively. Therefore, mixed matrix membrane (MMM) can be a solution to overcome the limitations of the two types of membranes by combining the good $CO_2/CH_4$ and $H_2/CH_4$ separation characteristics of inorganic materials and the desired mechanical properties of polymer membranes [17,23].

We have recently developed a new type of filler on a MMM, namely zeolite templated carbon (ZTC). ZTC is produced by removing the zeolite template of zeolite composite carbon (ZCC), as described in our study concerning ZCC fillers [16,17,24–26]. In the MMM PSF/ZTC, polysulfone acts as the polymeric matrix and ZTC as the filler [27,28]. Polysulfone is a type of glassy polymer, which is rigid and has better selectivity than rubbery polymer [13,26]. This improved selectivity occurs because the gas separation performance of polysulfone depends on differences in the size or kinetic diameter of the gas. By contrast, the gas separation performance of the rubbery membrane is based on condensation [29]. ZTC uses zeolite-Y as a hard template to produce a high microporosity, which has the capacity to adsorb a large amount of $CO_2$ [30]. Gunawan *et al*. [30] synthesized ZTC, which has an excellent adsorption capacity of $CO_2$, namely $9.51 \pm 0.48$ wt%, and is able to desorb $CO_2$ (77.5%). Possessing these fascinating properties showed that ZTC potentially could be applied as the filler in MMM. Our results showed that the presence of ZTC as the filler in the MMM-based polysulfone increased the selectivity of $CO_2/CH_4$ from 2.56 to 9.99 and selectivity of $H_2/CH_4$ from 7.77 to 28.88 [27]. The other fillers applied in MMM-based polysulfone, such as zeolite and silica, have been reported by several researchers. Mohamat *et al*. [31] reported that the incorporation of 3wt% zeolite T in polysulfone membrane enhanced $CO_2/CH_4$ selectivity from 2.63 to 3.37, with $CO_2$ permeability of 78.90 GPU. On the other hand, the presence of 2 wt% of KIT-6 (KIT: Korea Advanced Institute of Science and Technology), a silica mesoporous, could improve $CO_2/CH_4$ selectivity to 32.4, with $CO_2$ permeability of 5.4 Barrer [32]. However, the disadvantage of MMM is the weak interaction between the polymer matrix and the filler, which can form voids [33] and thereby reduce the separation performance. To the best of our knowledge [27,34], the nature of carbon makes it incompatible with the organic polymer matrix, which leads to poor interfacial adhesion. Therefore, modifications are needed to improve the separation performance of the membrane.

Annealing is one of the easiest and most economical methods to increase the interaction between polymer and filler. Annealing the MMM can make polymer chains more flexible and interact better with inorganic filler [35]. Annealing the membrane at a temperature higher than its glass transition temperature (Tg) can also result in better polymer chain bonding with the filler [36]. Another technique that can improve membrane performance is coating. In the research of Ismail et al. [37], MMM PES/zeolite coated with Dynasylan Ameo (DA) 10 %wt increased the selectivity of $CO_2/CH_4$ from 2.86 to 15.43. The coating on the membrane covers the voids because the coating material increases the adhesion between the polymer matrix and filler particles. Moreover, membrane coating can also increase thermal and chemical stability, as well as selectivity of membranes. For example, silane coating enhanced the Tg of the MMM by about 1–4°C (from 219.05 to 224.51°C) followed by use of higher silane concentrations [37]. Recently, tetramethylorthosilicate (TMOS) deposition has been studied by Nomura et al. [38,39], in which $H_2$ permeance was found to be approximately $2 \times 10^{-7}$ mol m$^{-2}$ s$^{-1}$ Pa$^{-1}$. Compared with other kinds of silane, TMOS offers advantages such as a smooth membrane surface, dense silica, and the lowest activation energy for $H_2$ permeation (10.5 kJ mol$^{-1}$) [38]. Generally, TMOS is used for silica membrane preparation, while there are no reports of coating membranes using TMOS on MMM. The most common coating material that is applied to MMM is polydimethylsiloxane [17,40–44]. On the other hand, the selection size of the silane coating agent influences the gas diffusion compatibility. For example, poly(N-vinylpyrrolidone) is not suitable for surface modification of microporous inorganic fillers since the polymer sizing on the particle surface is prone to cause pore blockage [45]. Thus, this study examined another potential coating material, namely TMOS, which has unique properties to enhance gas separation performance.

This research is a continuation of a previous study [27], which focused on the post-treatment of interfacial voids to improve the gas separation performance of MMM PSF/ZTC by modification via annealing, coating, and both combinations. Annealing was carried out at temperatures of 120, 150, and 190°C. These temperatures are above and below the Tg of polysulfone (186°C) to elucidate polymer matrix densification due to the annealing process. On the other hand, coating was carried out with variations in the concentration of TMOS of 0.01, 0.03, and 0.05 mol. In addition, a combination of annealing at 190°C and coating with various concentrations of TMOS was also carried out, which is illustrated in figure 1. These parameters were used to comprehensively investigate the character and performance of each modification.

# 2. Materials and methods

## 2.1. Materials

The materials used in this study were divided into four stages, namely: (1) the materials used for ZTC synthesis were zeolite-Y template (Na-form, HSZ320NAA) supplied by Tosoh, furfuryl alcohol (FA), mesitylene, propylene gas (4% in $N_2$), and fluoric acid (HF, 46%, purchased from Merck); (2) the materials used for the preparation of the MMM PSF/ZTC were N,N-dimethylacetamide (DMAc, 99%, provided by Merck), tetrahydrofuran (THF, 99.8%, supplied by QreC), polysulfone (Udel-P3500) supplied by Amoco Chemicals (USA), ethanol (EtOH) provided by Merck, N-methyl-2-pyrrolidone (NMP) purchased from QreC, distilled water, and methanol (MeOH, 99.9%, procured from Merck); (3) the materials used for the modification of the MMM PSF/ZTC with variations in the concentration of the coating material were TMOS (Shin-Etsu), n-hexane ($C_6H_{14}$ 95%, Kanto), and purified water; (4) the materials used for the gas permeation test were cotton, epoxy resin, MMM PSF/ZTC, ultra-high purity $CO_2$ gas (99.99%), ultra-high purity $H_2$ gas (99.99%), and ultra-high purity $CH_4$ gas (99.99%). The reasons for the selection of these materials are described in table 1.

## 2.2. Methods

### 2.2.1. Preparation of zeolite templated carbon (ZTC)

The preparation of ZTC was the same as in the previous study [30]. Zeolite-Y channels were first impregnated with furfuryl alcohol (FA) using the chemical vapour deposition (CVD) method. The dried zeolite-Y was placed in a flask and dried at 200°C under vacuum for 6 h. Liquid FA was then put into the flask under reduced pressure. Then, the pressure was returned to atmospheric pressure by flowing $N_2$ into the system. The mixture was stirred at room temperature for 3 h and subsequently filtered, followed by washing with mesitylene to remove residual FA on the external zeolite surface. The washing process was

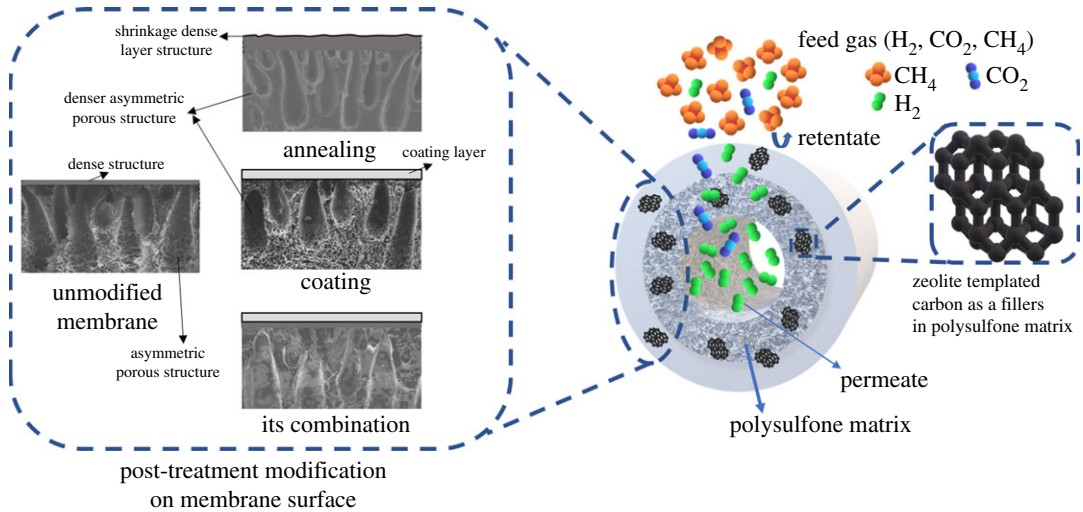

**Figure 1.** Illustration of the modification of post-treatment hollow fibre PSF/ZTC MMM.

**Table 1.** Names and structures of substances chosen for the membrane, along with reasons for their selection.

| chemical | reason for selection | structure |
|---|---|---|
| Polysulfone [28] | high intrinsic selectivity<br>good mechanical and thermal properties<br>ease of fabrication | |
| zeolite templated carbon [27] | well-defined pore structure<br>a large surface area<br>good pore structure with no rigid pore properties<br>possesses a negative replica of the zeolite-Y structure | |

repeated three times. The FA polymerization was conducted by heating at 150°C for 24 h under a $N_2$ flow. The obtained zeolite/PFA composite was heated at 700°C for 2 h to carbonize PFA in the zeolite channels. Then, propylene gas (4% in $N_2$) was passed through the reactor and held for 2 h. The thermal decomposition of propylene resulted in pyrolytic carbon deposition in the zeolite channels. The prepared zeolite/carbon composite was further heat-treated at 900°C for 3 h under a $N_2$ flow, with the resultant material being ZCC. The zeolite framework in the composite sample was dissolved by washing with an excess amount of 46% aqueous HF solution at room temperature for 5 h. The sample was then filtered and washed with pure water three times, followed by drying. The final product was then obtained, namely ZTC.

### 2.2.2. Membrane preparation

ZTC at a concentration of 0.25 wt% was suspended in 30 g DMAc via sonication. To achieve better dispersion, the suspension was further sonicated with a Q125 micro-tip sonicator (amplitude 100%, 2 s elapsed time). 30 g of THF was added into the suspension and placed in an ultrasonic bath for the 10 min for the sonication process. 10 g of PSF was then gradually added to the solvent mixture three times and stirred until the solution was homogeneous. About 10 g ethanol was added via drops into the solution and vigorously stirred. Finally, the resulting mixture was sonicated in an ultrasonic water bath for 1 h and left for 24 h at room temperature to remove microbubbles. The MMMs were fabricated by a dry-jet wet-spinning process. The dope solution reservoir was connected to a spinneret with outer/inner diameter dimensions of 0.8 mm/0.4 mm by a gear pump. The dope solution flow rate was set at 1 ml min⁻¹. Bore coagulant containing 90 vol% NMP and 10 vol% distilled water was simultaneously connected to the

spinneret by a syringe pump at a flow rate of 0.7 ml min$^{-1}$. The fibres were then extruded from the spinneret and guided into a coagulation bath of water. The dry gap distance between the water and the spinneret was controlled at 4 cm. The hollow fibres were then collected by a wind-up drum at take-up speed of 10 m min$^{-1}$. The obtained fibres were cut and immersed in another water bath for 48 h to remove excess solvent, with the water being replaced several times. The fibres were then post-treated in methanol for 4 h to reduce pore collapse and shrinkage during the drying process at room conditions for 48 h. The membrane preparation was adopted from previous literature [27,28].

### 2.2.3. Post-treatment of membrane

Post-treatment was conducted with various methods, explained as follows. MMM PSF/ZTC was annealed using a muffle furnace under vacuum conditions with a heating rate of 0.3°C/min and a holding time of 1 h. Annealing was done at temperatures of 120, 150, and 190°C. On the other hand, TMOS solution was prepared by mixing TMOS and 132 g $n$-hexane. The coating was performed at varying concentrations of 0.01, 0.03, and 0.05 mol. Five membrane fibres were placed into the solution, then stirred for 10 min. After the stirring process was completed, the solution containing PSF membranes was refluxed at 60°C for 2 h, accompanied by stirring. The membranes were then dried at room temperature for 24 h.

Additionally, MMM PSF/ZTC, which had been annealed using a muffle furnace at 190°C, was then coated with TMOS solution at varying concentrations of 0.01, 0.03, and 0.05 mol. The conditions and methods used for the individual annealing and coating treatments were the same in the case of the combination of both treatments.

### 2.2.4. Membrane characterization

MMM PSF/ZTC was characterized via X-ray diffraction (XRD SmartLab, Rigaku) to identify changes in crystal structure and intermolecular distances between polymer intersegmental chains during the process of annealing. The functional groups of the membranes before and after coating treatment were also analysed using attenuated total reflection-Fourier transform infrared (ATR-FTIR IRAffinity-1S, Shimadzu). The morphology of MMM PSF/ZTC was characterized via scanning electron microscopy (SEM, Keyence VE-8800) and field emission scanning electron microscopy (FESEM JSM-7610F, JEOL) to observe the compatibility between particles and polymer matrix. Thermal gravimetric analysis (TGA-50, Shimadzu) was used to determine the thermal stability of MMM PSF/ZTC, which was based on the reduction in mass that occurred in the membrane.

### 2.2.5. Gas permeation test

The single gas permeation test is described as follows. $CO_2$, $H_2$, and $CH_4$ permeation was carried out by bubble flow and pressure difference methods. Both methods have been explained in more detail in Myagmarjav *et al*. [46]. Furthermore, the binary gas test was carried out using the MMM PSF/ZTC, which had been tested for single gas. Gas permeation measurements were carried out using gas $CO_2/CH_4$ (50/50%) and $H_2/CH_4$ (50/50%) at room temperature with a pressure of 2 bar. The gas composition in the permeate was analysed using gas chromatography (GC-8A TCD and GC-2014 FID, Shimadzu). The gas permeation rig is illustrated in figure 2.

The permeance value can be obtained through equation (2.1) [47] :

$$P_i = \frac{n_i \; x \; l}{t \; A \; \Delta P} \qquad (2.1)$$

where $P_i$ is the gas permeation in mol s$^{-1}$ m$^{-2}$ Pa$^{-1}$ (1 GPU = 3.35 × 10$^{-10}$ mol s$^{-1}$ m$^{-2}$ Pa$^{-1}$), $n_i$ [mol] is the permeated molecules, $t$ [s] is the permeation time, $\Delta P$ [Pa] is the pressure differential, $l$ is the thickness of the membrane (m), and $A$ is the effective membrane surface area (m$^2$).

# 3. Results and discussion

## 3.1. Annealing treatment of membrane

The gas separation performance of MMM PSF/ZTC before and after annealing can be seen from the permeation and selectivity values, which are shown in figure 3. MMM PSF/ZTC without annealing

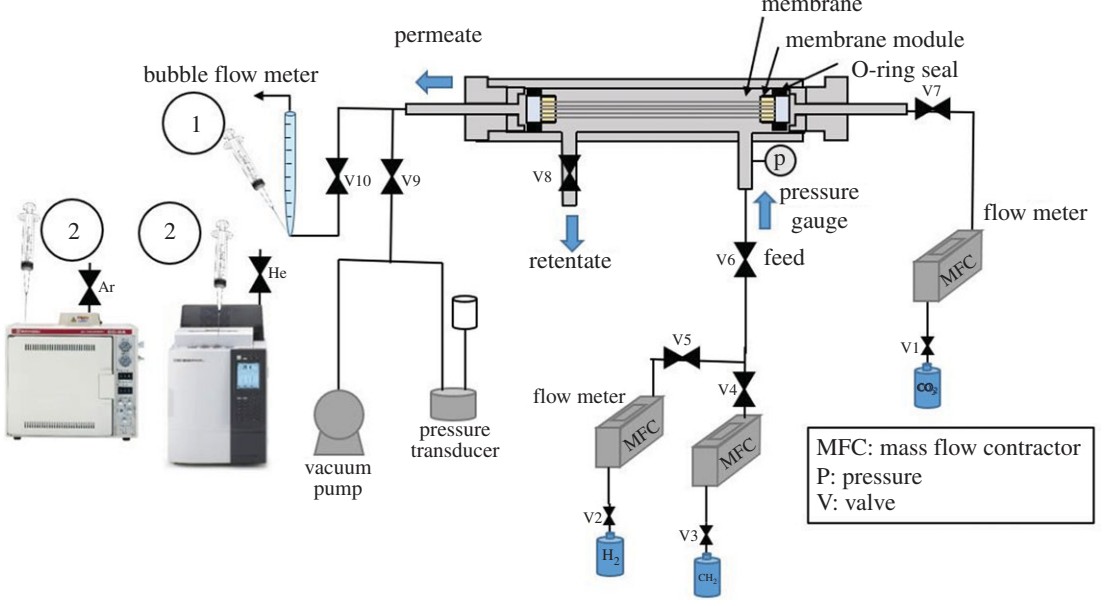

**Figure 2.** Schematic diagram of the gas permeation rig.

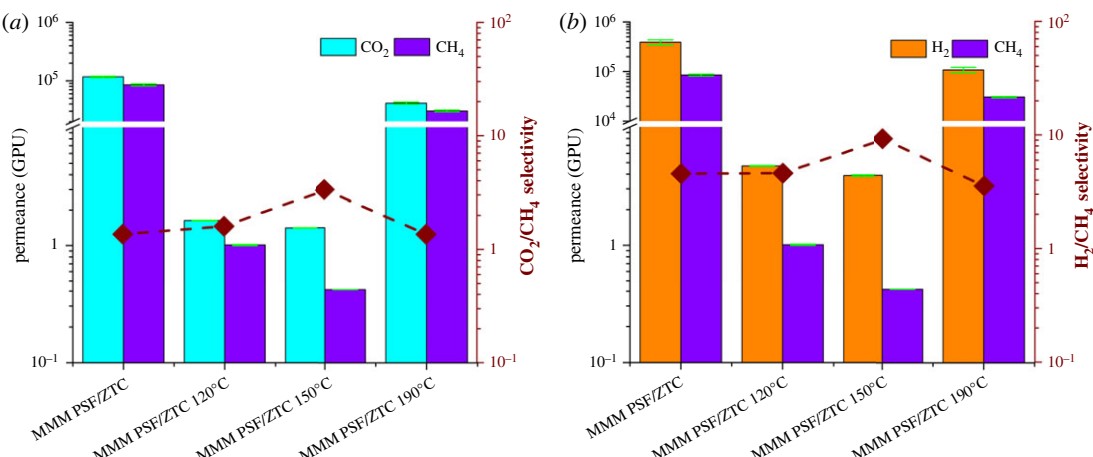

**Figure 3.** Permeation and selectivity of (*a*) $CO_2/CH_4$ and (*b*) $H_2/CH_4$ on MMM PSF/ZTC with variations of annealing temperature.

had high permeation but the selectivity was low, as shown in table 2. The $CO_2/CH_4$ and $H_2/CH_4$ selectivity of MMM PSF/ZTC without annealing exceeds that of Knudsen due to the presence of micropore ZTCs with an average pore size of 1.21 nm, which was larger than ultramicropores (<0.6 nm) [48]. The pore size was larger than the diameter of the $CH_4$, $CO_2$, and $H_2$ gas molecules; thus, the gas transport mechanism did not follow molecular sieving. Therefore, the possible gas transport mechanism on MMM PSF/ZTC without annealing was the surface flux mechanism through the micro and meso pores of ZTC, where the surface diffusion mechanism was suitable for fast gases ($H_2$) and gases with larger kinetic diameters ($CO_2$ and $CH_4$) diffused slowly on ZTC micropores [49].

According to the SEM observation, the finger-like pore was formed during the dry/wet-spinning process as a consequence of phase inversion between the coagulation liquid and polymer solution. On the other hand, the presence of voids in MMM PSF/ZTC without annealing, as shown in Wijiyanti *et al*. [27], was due to the low adhesion between the polymer matrix and the ZTC. This encouraged the modification of the membrane by annealing to improve the gas separation performance. MMM PSF/ZTC annealed at 120 and 150°C had a reduction in permeation of 99% for all gases. The reduction in permeation was inversely related to selectivity. The increases in selectivity were 0.97% for $H_2/CH_4$ and 17% for $CO_2/CH_4$ gas at an annealing temperature of 120°C, and 102% for $H_2/CH_4$ and 144% for $CO_2/CH_4$ gas at an annealing temperature of 150°C. This was due to the changes in the

**Table 2.** Permeation and selectivity of single gases on MMM PSF/ZTC with variations of annealing temperature, coating treatment, and a combination of both treatments.

| membrane | permeation (GPU) | | | | selectivity | |
| | $H_2$ | $CO_2$ | $CH_4$ | | $H_2/CH_4$ | $CO_2/CH_4$ |
| --- | --- | --- | --- | --- | --- | --- |
| MMM PSF/ZTC | $389\,775.29 \pm 46\,321.49$ | $116\,361.03 \pm 3477.65$ | $85\,088.91 \pm 3392.87$ | | 4.58 | 1.37 |
| MMM PSF/ZTC annealed at 120°C | $4.69 \pm 0.041$ (−99%) | $1.63 \pm 0.007$ (−99%) | $1.01 \pm 0.013$ (−99%) | | 4.63 (0.97%) | 1.61 (17%) |
| MMM PSF/ZTC annealed at 150°C | $3.89 \pm 0.054$ (−99%) | $1.41 \pm 0.003$ (−99%) | $0.42 \pm 0.001$ (−99%) | | 9.26 (102%) | 3.35 (144%) |
| MMM PSF/ZTC annealed at 190°C | $107\,433.20 \pm 13\,317.84$ (−72%) | $41\,229.82 \pm 1683.41$ (−64%) | $30\,293.33 \pm 1033.68$ (−64%) | | 3.55 (−22%) | 1.36 (−0.48%) |
| MMM PSF/ZTC coated with 0.01 mol TMOS | $13\,818.57 \pm 420.95$ (−96%) | $2887.87 \pm 36.64$ (−97%) | $5297.68 \pm 58.03$ (−93%) | | 2.61 (−43%) | 0.55 (−60%) |
| MMM PSF/ZTC coated with 0.03 mol TMOS | $444.11 \pm 7.76$ (−99%) | $5.42 \pm 0.10$ (−99%) | $6.75 \pm 0.10$ (−99%) | | 65.76 (1335.52%) | 0.80 (−41%) |
| MMM PSF/ZTC coated with 0.05 mol TMOS | $423.67 \pm 17.83$ (−99%) | $157.35 \pm 2.27$ (−99%) | $125.88 \pm 2.57$ (−99%) | | 3.37 (−26%) | 1.25 (−8%) |
| MMM PSF/ZTC 190°C 0.01 mol TMOS | $128.08 \pm 1.80$ (−99%) | $58.55 \pm 4.39$ (−99%) | $58.47 \pm 1.70$ (−99%) | | 2.19 (−52%) | 1.00 (−26%) |
| MMM PSF/ZTC 190°C 0.03 mol TMOS | $0.02 \pm 0.0002$ (−99%) | $0.01 \pm 0.0001$ (−99%) | $0.002 \pm 0.0001$ (−100%) | | 10.83 (136%) | 5.90 (331%) |
| MMM PSF/ZTC 190°C 0.05 mol TMOS | $0.11 \pm 0.0006$ (−99%) | $0.037 \pm 0.0002$ (−99%) | $0.040 \pm 0.0007$ (−100%) | | 2.79 (−39%) | 0.93 (−32%) |
| Knudsen selectivity | | | | | 2.83 | 0.6 |

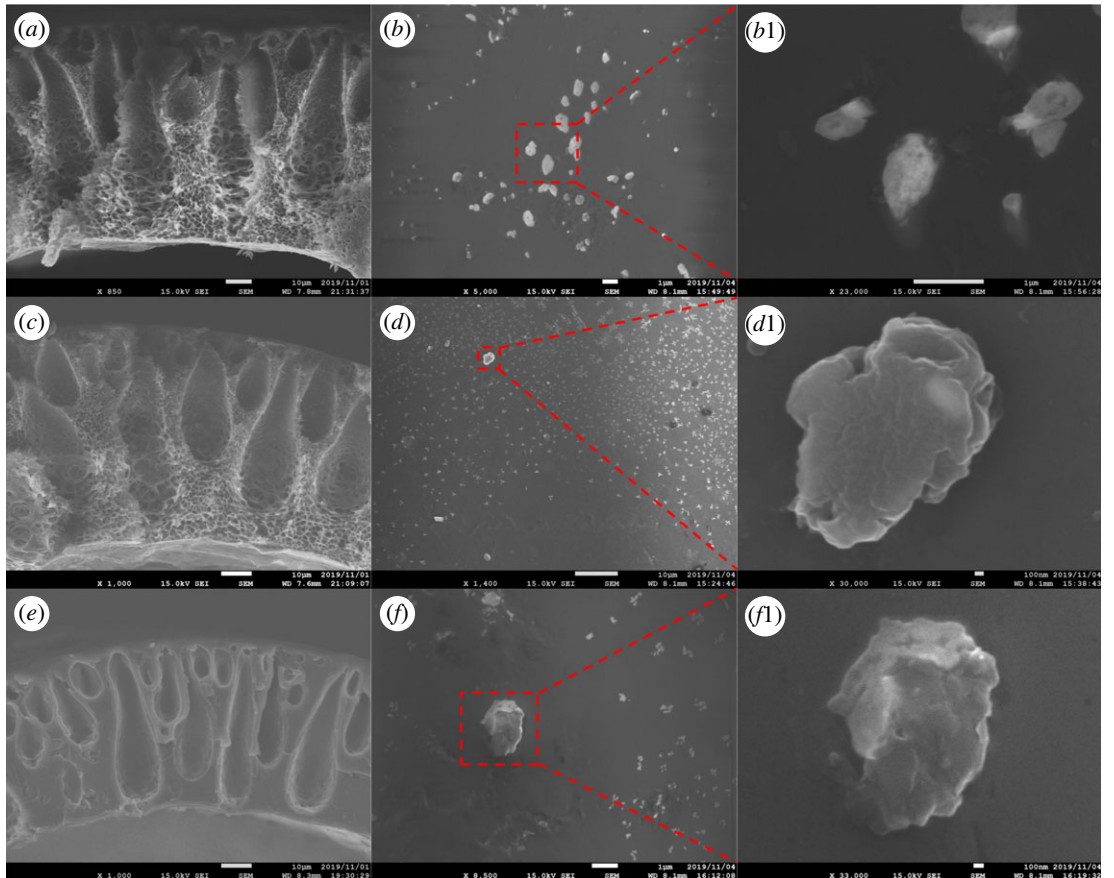

**Figure 4.** SEM morphology of MMM PSF/ZTC's cross-section annealed at (*a*) 120°C, (*c*) 150°C, and (*e*) 190°C, and its surfaces annealed at (*b*) 120°C, (*d*) 150°C, and (*f*) 190°C (labels suffixed with "1" are close-ups).

structure of the polymer matrix, which became a denser and smoother surface, as supported by the cross-section and surface morphology of SEM in figure 4*a–d* compared with the membrane without annealing. The shrinkage of the pores caused gas with a large molecular diameter to be more difficult to diffuse in the polymer chain. This is in accordance with the research of Jiang *et al.* [50], who heated MMM PSF/zeolite *β* at temperatures of 120 and 150°C.

Unlike the case of MMM PSF/ZTC annealed at 190°C (above the Tg of polysulfone of 186°C), the resulting gas permeation decreased, but not by more than the membranes annealed at 120 and 150°C, which was 64% for $CH_4$ and $CO_2$ gas, and 72% for $H_2$ gas. The insignificant reduction in gas permeation on the membrane was caused by the polymer structure becoming denser and rubbery, as seen in the cross-section and surface morphology (figure 4*e,f*). In addition, MMM PSF/ZTC annealed at 190°C also had a reduced selectivity of 22% for $H_2/CH_4$ and 0.48% for $CO_2/CH_4$ gas.

In addition to using SEM analysis, structural changes in the membrane can also be reviewed by XRD analysis. Annealed MMM PSF/ZTC had a typical PSF diffraction peak shift, as shown in table 3. MMM PSF/ZTC without annealing and MMM PSF/ZTC annealed at 120, 150, and 190°C had a wide peak PSF at $2\theta = 17.88°$ (*d*-spacing = 0.500 nm); $2\theta = 24.19°$ (*d*-spacing = 0.368 nm); 23.70° (*d*-spacing = 0.375 nm); 23.15° (*d*-spacing = 0.384 nm), respectively (figure 5*a*). Annealing at 120°C can reduce the *d*-spacing of MMM PSF/ZTC, which was caused by the denser polymer matrix, so that the mobility of the polymer chains becomes smaller [51]. On MMM PSF/ZTC annealed at 150 and 190°C, the *d*-spacing had increased compared with MMM PSF/ZTC annealed at 120°C, which indicated a change in the properties of the polymer chain to become more flexible.

On the other hand, the denser membrane structure due to annealing can increase thermal stability [51], which can be seen through TGA analysis (figure 5*b*). This was indicated by a change in decomposition temperature (table 4) of MMM PSF/ZTC annealed at 120, 150, and 190°C compared with the membrane without heating. MMM PSF/ZTC without annealing had a decomposition temperature of 493.17°C, while the decomposition temperatures of MMM PSF/ZTC annealed at 120, 150, and 190°C were 505.12, 515.2, and 516.9°C, respectively. The thermal stability enhancement was

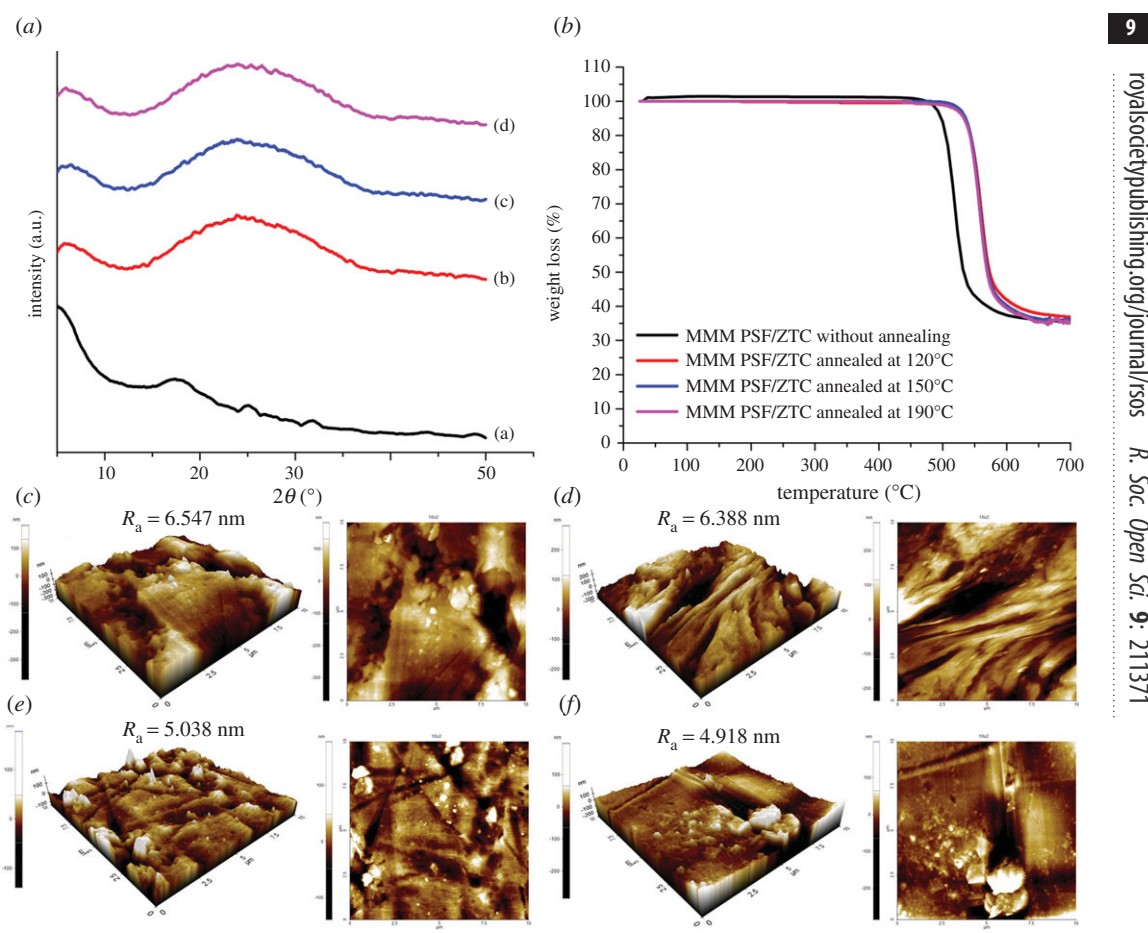

**Figure 5.** MMM PSF/ZTC characterizations of (*a*) XRD diffractogram ((*a*) without annealing, and annealed at (*b*) 120°C, (*c*) 150°C, and (*d*) 190°C); (*b*) TGA curve; and AFM images (*c*) without modification, and annealed at (*d*) 120°C, (*e*)150°C, and (*f*) 190°C.

**Table 3.** XRD parameters for MMM PSF/ZTC.

| membrane | 2θ PSF | *d*-spacing (nm) PSF |
|---|---|---|
| MMM PSF/ZTC without annealing | 17.88 | 0.500 |
| MMM PSF/ZTC annealed at 120°C | 24.19 | 0.368 |
| MMM PSF/ZTC annealed at 150°C | 23.70 | 0.375 |
| MMM PSF/ZTC annealed at 190°C | 23.15 | 0.384 |

due to the higher adhesion of the polymer and ZTC as a result of heating at high temperatures [52]. The increase in adhesion itself appeared as a result of increased chain mobility leading to an increase in affinity to the solid surface. Annealed MMM PSF/ZTC had better thermal stability than MMM PSF/ZTC without annealing. The same result was shown in the research of Zhuang *et al*. [53], who heated the poly(2,6-dimethyl-1,4-phenylene oxide) (PPO)-silica MMMs at a temperature of 220°C and showed an increase in the decomposition temperature from 450.06 to 454.37°C. On the other hand, annealing improved the mechanical characteristics of polymeric membranes by encouraging stronger connections between polymer chains and a greater degree of crystallinity in the polymer matrix. Annealing the membrane at 100, 150, or 200°C enhanced its mechanical strength from 372 to 586, 734, or 743 MPa, respectively [54].

Atomic force microscopy (AFM) observations examined the external surface topographies of MMM PSF/ZTC with various treatments. Figure 5*c–f* presents the plane and three-dimensional topography of MMM PSF/ZTC without modification, and annealing at 120, 150, and 190°C. According to the AFM data, increasing the annealing temperature contributed to decreasing the average roughness ($R_a$) of the

**Table 4.** Decomposition temperature of MMM PSF/ZTC.

| membrane | $T_d$ (°C) | weight loss (%) |
|---|---|---|
| MMM PSF/ZTC without annealing | 493.17–554.83 | 57.190 |
| MMM PSF/ZTC annealed at 120°C | 505.12–569.23 | 54.110 |
| MMM PSF/ZTC annealed at 150°C | 515.2–566.12 | 55.850 |
| MMM PSF/ZTC annealed at 190°C | 516.9–564.14 | 57.200 |

membrane surface. Similarly, Barzin *et al.* [55] reported the decreasing roughness of the membrane followed by improving selectivity. Supporting the gas separation performance, the selectivity of $CO_2$/ $CH_4$ and $H_2$/$CH_4$ was enhanced on membranes annealed at 120 and 150°C due to the smoother surface. The greatest decrease in the membrane roughness after annealing at 190°C did not influence the selectivity because the membrane had turned to a rubbery polymer.

In addition, the gas performance of MMM PSF/ZTC was also tested using binary gas (50/50% $CO_2$/ $CH_4$ and 50/50% $H_2$/$CH_4$) at room temperature with a pressure of 2 bar. $CO_2$/$CH_4$ gas pair testing was carried out using all types of membranes, while $H_2$/$CH_4$ gas pair testing was only carried out using MMM PSF/ZTC annealed at 190°C, as shown in table 5. The results of the binary gas separation performance obtained were different from the results of the single gas separation performance, where the permeation and selectivity of the binary gas decreased. At the same gas pressure, the expected result was close to the single gas separation performance. The reduction in gas separation performance was due to the competition between $CO_2$ or $H_2$ gas and $CH_4$ gas on the absorption side of the membrane [56]. In the study of Kim *et al.* [57], the reduction in $H_2$ permeation was higher than $CH_4$ in the $H_2$/$CH_4$ gas pair because $H_2$ gas had a lower critical temperature at 33.2 K compared with $CH_4$ gas (190.55 K). This caused the absorption of $CH_4$ gas to be greater than $H_2$. Consequently, the absorption of $CH_4$ on the polymer matrix competitively reduced the absorption of $H_2$. In the $CO_2$/ $CH_4$ gas pair, the higher critical temperature of $CO_2$ (304.25 K) can reduce the absorption of $CH_4$. In addition, the rapid diffusion of $CO_2$ on the membrane will facilitate the diffusion of $CH_4$ gas [56]. The increase in $CH_4$ diffusion, which was much greater than the reduction in $CH_4$ absorption, caused a smaller reduction, or even an increase, in $CH_4$ permeation than $CO_2$. Thus, the selectivity of the $CO_2$/$CH_4$ gas mixture was lower than the ideal selectivity.

## 3.2. Coating treatment of membrane

Another method that can be used to improve the gas separation performance on the MMM PSF/ZTC is coating using TMOS with a variation of the concentration of 0.01, 0.03, and 0.05 mol. The gas separation performance results are shown in figure 6 and table 2.

In the research of Ismail *et al.* [37], coating using silane can increase the selectivity of gas separation. However, this present study was different. In the case of coated MMM PSF/ZTC, $H_2$, $CO_2$, and $CH_4$ permeance was less than MMM PSF/ZTC without coating. This was due to the formation of a coating layer that enhances gas transport resistance [58]. Furthermore, the reduction in $CO_2$/$CH_4$ and $H_2$/$CH_4$ selectivity occurred on MMM PSF/ZTC coated with 0.01 mol TMOS because membrane surfaces tend to be covered by high TMOS concentrations with tighter polymer chain packaging, which causes a reduction in free volume.

On MMM PSF/ZTC coated with 0.03 mol TMOS, there was a reduction in $H_2$, $CO_2$, and $CH_4$ permeation accompanied by a reduction in $CO_2$/$CH_4$ selectivity and an increase in $H_2$/$CH_4$ selectivity. The increase in $H_2$/$CH_4$ selectivity occurred because TMOS reduces the activation energy for $H_2$ gas permeation [59]. When compared with MMM PSF/ZTC coated with 0.01 mol TMOS, the increase in $CO_2$/$CH_4$ and $H_2$/$CH_4$ selectivity on MMM PSF/ZTC coated with 0.03 mol TMOS was due to the higher concentration of TMOS on the membrane, which reduced the micro void around the filler more optimally. This was in accordance with the research of Ismail *et al.* [37]. It suggests that 0.03 mol TMOS utilization could disturb the diffusion of larger-sized gas molecules (i.e. $CO_2$ and $CH_4$) due to the membrane pores getting narrower, while $H_2$ penetration was not affected considerably. As seen in figure 7a–c, the coated MMM pore size was smaller than that of uncoated MMM. In addition, coating using TMOS produced a smooth membrane surface without any voids. As shown in the AFM topography, membrane coated with TMOS reduces the surface roughness from 6.547 to 5.756 nm (figure 7d).

**Table 5.** Permeation and selectivity of binary gas on MMM PSF/ZTC with variations of annealing temperature, coating treatment, and a combination of both treatments.

| membrane | binary gas permeation (GPU) | | binary gas selectivity | binary gas permeation (GPU) | | binary gas selectivity | | ideal selectivity | |
| --- | --- | --- | --- | --- | --- | --- | --- | --- | --- |
| | $CO_2$ | $CH_4$ | $CO_2/CH_4$ | $H_2$ | $CH_4$ | $H_2/CH_4$ | | $CO_2/CH_4$ | $H_2/CH_4$ |
| MMM PSF/ZTC | 38.67 | 40.78 | 0.95 | — | — | — | | 1.37 | 4.58 |
| MMM PSF/ZTC annealed at 120°C | 99.60 | 55.74 | 1.79 | — | — | — | | 1.61 | 4.63 |
| MMM PSF/ZTC annealed at 150°C | 74.57 | 25.34 | 2.94 | — | — | — | | 3.35 | 9.26 |
| MMM PSF/ZTC annealed at 190°C | 5.60 | 19.42 | 0.29 | 0.42 | 0.44 | 0.96 | | 1.36 | 3.55 |
| MMM PSF/ZTC coated with 0.01 mol TMOS | — | — | — | — | — | — | | 0.55 | 2.61 |
| MMM PSF/ZTC coated with 0.03 mol TMOS | 44.34 | 54.32 | 0.82 | 159.73 | 69.99 | 2.28 | | 0.80 | 65.76 |
| MMM PSF/ZTC coated with 0.05 mol TMOS | — | — | — | — | — | — | | 1.25 | 3.37 |
| MMM PSF/ZTC 190°C 0.01 mol TMOS | 14.27 | 25.56 | 0.56 | 4411.09 | 6312.42 | 0.69 | | 1.00 | 2.19 |

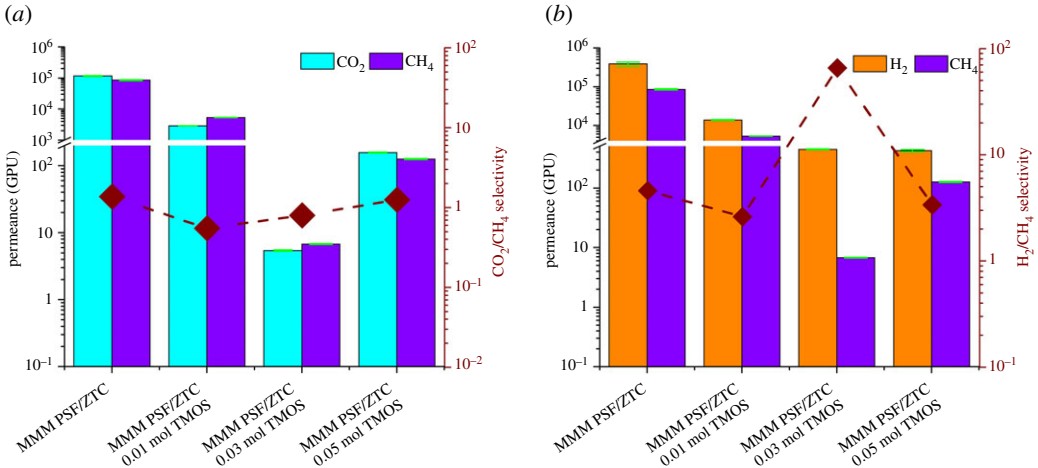

**Figure 6.** Permeation and selectivity of (a) $CO_2/CH_4$ and (b) $H_2/CH_4$ on MMM PSF/ZTC at various coating concentrations.

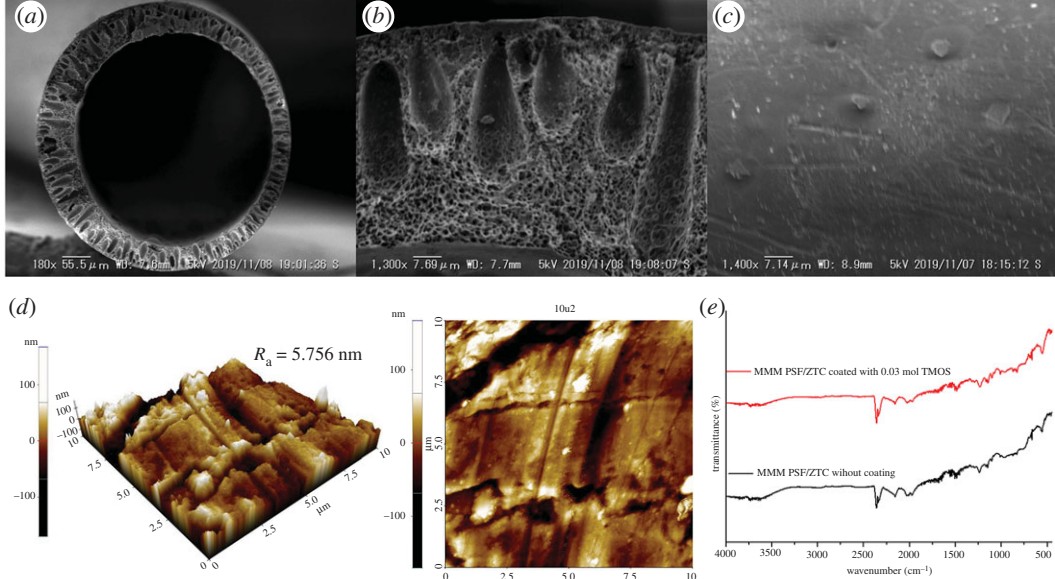

**Figure 7.** Coated MMM PSF/ZTC characterization of (a,b) cross-section and (c) its surface SEM images; (d) AFM images; and (e) FTIR spectra.

Decreased permeation and selectivity also occurred on MMM PSF/ZTC coated with 0.05 mol TMOS because the excess TMOS concentration on the membrane formed multilayers, which not only cover the voids but also block the gas diffusion path. This is in accordance with Roslan *et al.* [58], who stated that the viscosity of the solution increases with increasing Pebax concentration, which correlates with an enhancement in coating layer thickness. Furthermore, a higher Pebax coating concentration (9 wt%) on PSF membrane decreased $CO_2$ permeation to 11.55 GPU from 47.73 GPU (1 wt%) [58]. However, at 0.05 mol TMOS coating, a unique pattern was observed in which the penetration of $H_2$ gas decreased while the permeability of $CO_2$ and $CH_4$ gas increased. Due to the increase in concentration of the TMOS coating, it completely covers the membrane pores, favouring solution diffusion, which is more dependent on the gas's solubility. Moreover, it suggests that the incorporation of silane, which contains oxygen atoms, facilitates physical contact owing to its increased polarity [60,61]. Thus, increasing the TMOS concentration coating leads to an increase in $CO_2$ permeation that is greater than $CH_4$ permeation; as a result, $CO_2/CH_4$ selectivity improved. In addition, coating the membrane with TMOS resulted in no change to the molecular structure of the membrane, as shown by FTIR analysis (figure 7e). This indicates that there is no reaction between MMM PSF/ZTC and TMOS.

$CO_2/CH_4$ gas pair testing was carried out using MMM PSF/ZTC without coating and MMM PSF/ZTC coated with 0.03 mol TMOS, while $H_2/CH_4$ gas pair testing was only carried out using MMM PSF/

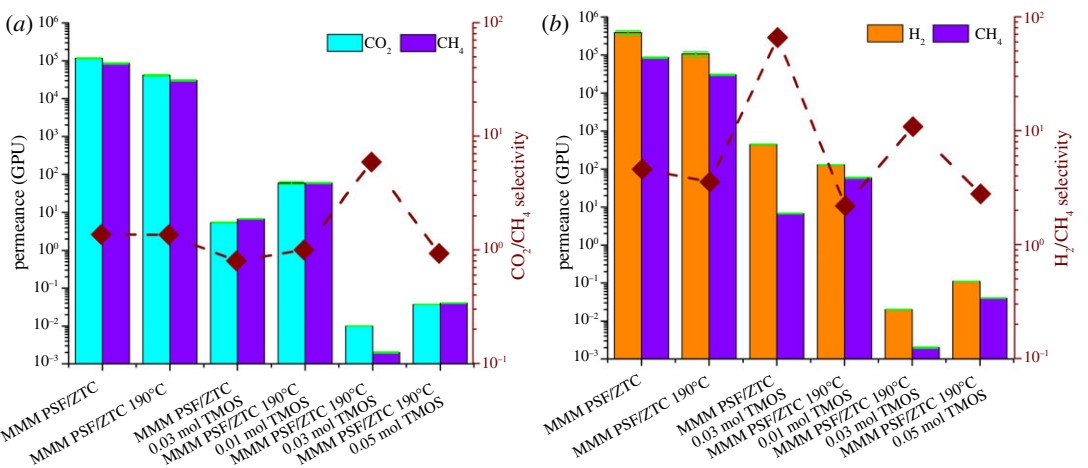

**Figure 8.** Permeation and selectivity of (*a*) $CO_2/CH_4$ and (*b*) $H_2/CH_4$ on MMM PSF/ZTC at 190°C with variations in TMOS concentration.

ZTC coated with 0.03 mol TMOS, as shown in table 5. The results of the binary gas separation performance obtained were different from the results of the single gas separation performance. The selectivity of the $CO_2/CH_4$ gas pair decreased in MMM PSF/ ZTC without coating and the increase in MMM PSF/ZTC coated with 0.03 mol TMOS was not significant. In the $H_2/CH_4$ gas pair, there was a significant reduction in selectivity. This was due to the existence of competition between gases, which has been explained in the discussion of the results of annealing MMM PSF/ZTC.

## 3.3. Combination of annealing and coating treatments

Compared with the other annealing temperatures, MMM PSF/ZTC annealed at 190°C exhibited the highest gas permeation performance. However, the rubbery and dense polymer structure on the membrane contributed to a low selectivity. Thus, the addition of post-treatment was carried out to try to increase the selectivity of the membrane. The gas separation performance comparison between membranes annealed at 190°C, coated with 0.03 mol TMOS, and the combination of annealing at 190°C and coating with various concentrations is shown in figure 8.

The MMM PSF/ZTC annealed at 190°C and coated with 0.01 mol TMOS reduced permeation and selectivity compared with the uncoated MMM PSF/ZTC annealed at 190°C, as shown in table 2. As discussed in the previous section, the reduction in permeation is due to the enhancement in gas transport resistance as the result of the formation of a coating layer [58]. Hypothetically, the pores of the membrane would become much smaller with the coating after annealing.

On MMM PSF/ZTC annealed at 190°C and coated with 0.03 mol TMOS, there was an increase in the $CO_2/CH_4$ and $H_2/CH_4$ selectivity compared with the uncoated MMM PSF/ZTC annealed at 190°C, because the higher concentration of TMOS in the membrane optimally reduced the micro void surrounding the filler [37]. This was supported by SEM analysis on the membrane surface (figure 9*a–c*), showing that the TMOS solution successfully covered the voids on MMM PSF/ZTC. The insignificant change in the morphology of the MMM PSF/ZTC annealed at 190°C before and after coating was due to the dense cross-section of the membrane as the result of annealing at 190°C and it was difficult to observe a difference. Furthermore, the AFM result exhibited that the MMM PSF/ZTC annealed at 190°C and coated with 0.03 mol TMOS was smoother than the membrane that had only been annealed (figure 9*d*). The reduction in average roughness impacts the permeability reduction. The external mass transfer influenced by surface morphology can create another complex layer in the process. The layer is essential for hindering the diffusion of larger gas molecules (like $CH_4$), which are affected by external mass transfer conditions via concentration polarization [62]. On the other hand, the excess concentration of TMOS on MMM PSF/ZTC annealed at 190°C and coated with 0.05 mol TMOS reduces permeation and selectivity because of the multilayer, which reduces the adhesion between the polymer matrix and filler particles [58]. Moreover, the selectivity (1.36) is lower than MMM PSF/ZTC annealed at 190°C. This suggests that membrane surfaces are typically covered by TMOS coating, resulting in tighter polymer chain packing and a decrease in free volume.

Testing of $CO_2/CH_4$ and $H_2/CH_4$ gases was carried out on membranes with a combination of annealing at 190°C and coating with TMOS (0.01 and 0.03 mol), as shown in table 5. The results of the binary gas separation performance were different from the results of the single gas separation performance, where

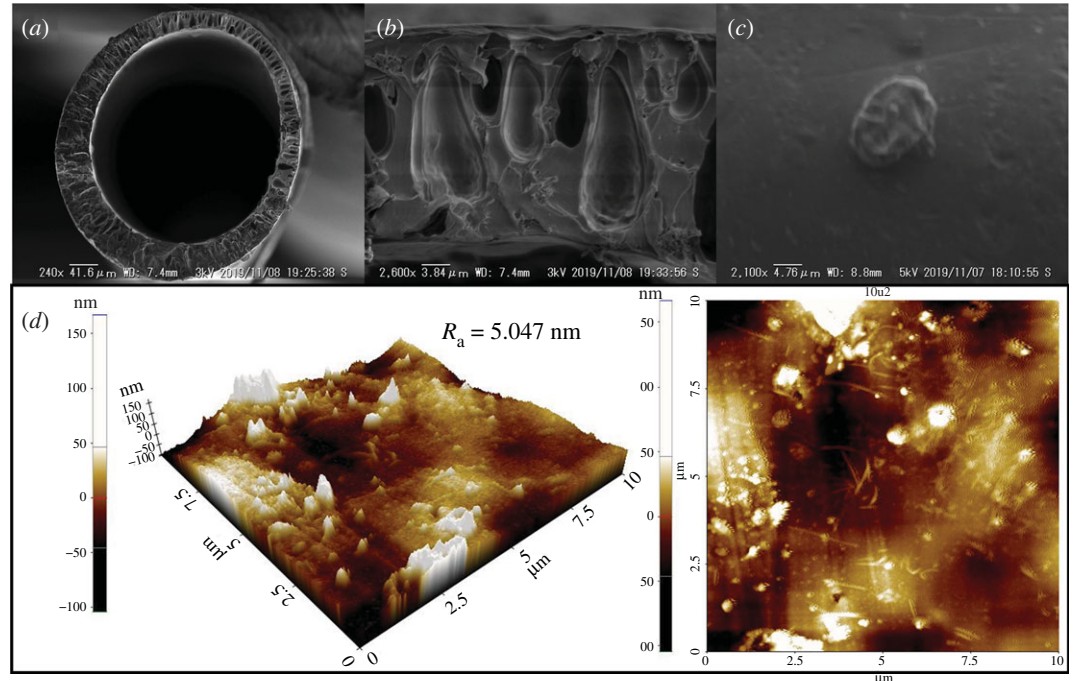

**Figure 9.** Double post-treatment (annealing and coating) on MMM PSF/ZTC characterization of (*a*,*b*) cross-section and (*c*) its surface SEM images; and (*d*) AFM images.

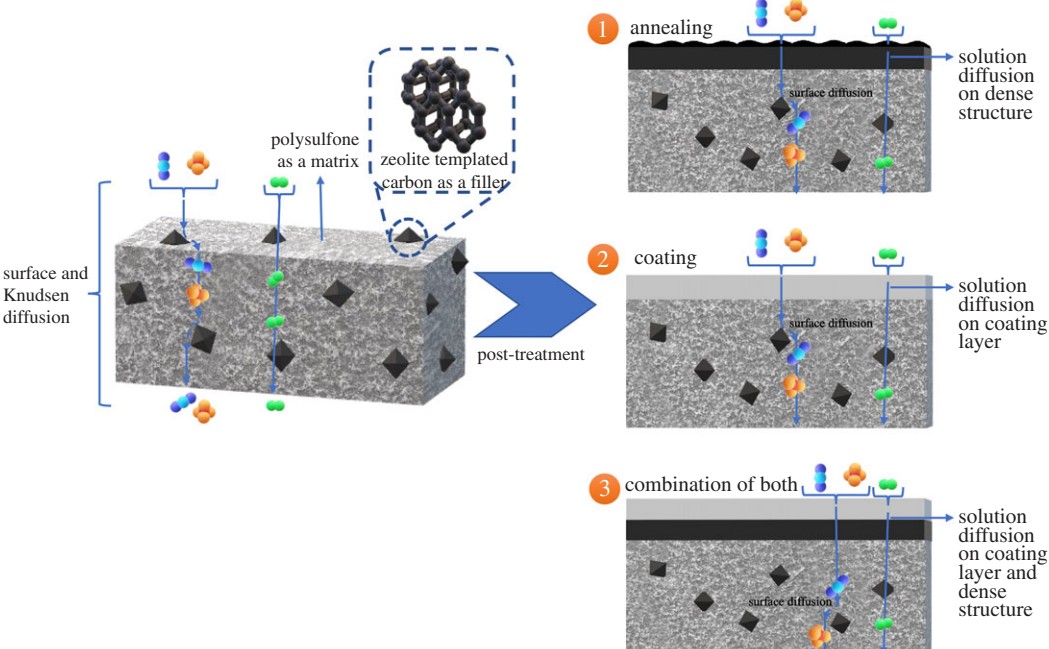

**Figure 10.** Prediction of the gas diffusion mechanism on MMM PSF/ZTC annealed at 190℃.

the $CO_2/CH_4$ and $H_2/CH_4$ selectivity were significantly reduced in the former. This was due to the existence of competition between gases, which has been explained in the previous discussion.

## 3.5. Overview of gas separation performance

The gas diffusion mechanism that occurred on MMM PSF/ZTC annealed at 120 and 150℃ was the same as the mechanism on membranes without annealing. This was different on MMM PSF/ZTC annealed at 190℃, which was rubbery, where the diffusion mechanism that played a role was a combination of solution diffusion and surface flux, as illustrated in figure 10. In this mechanism, $H_2$ gas tends to diffuse by

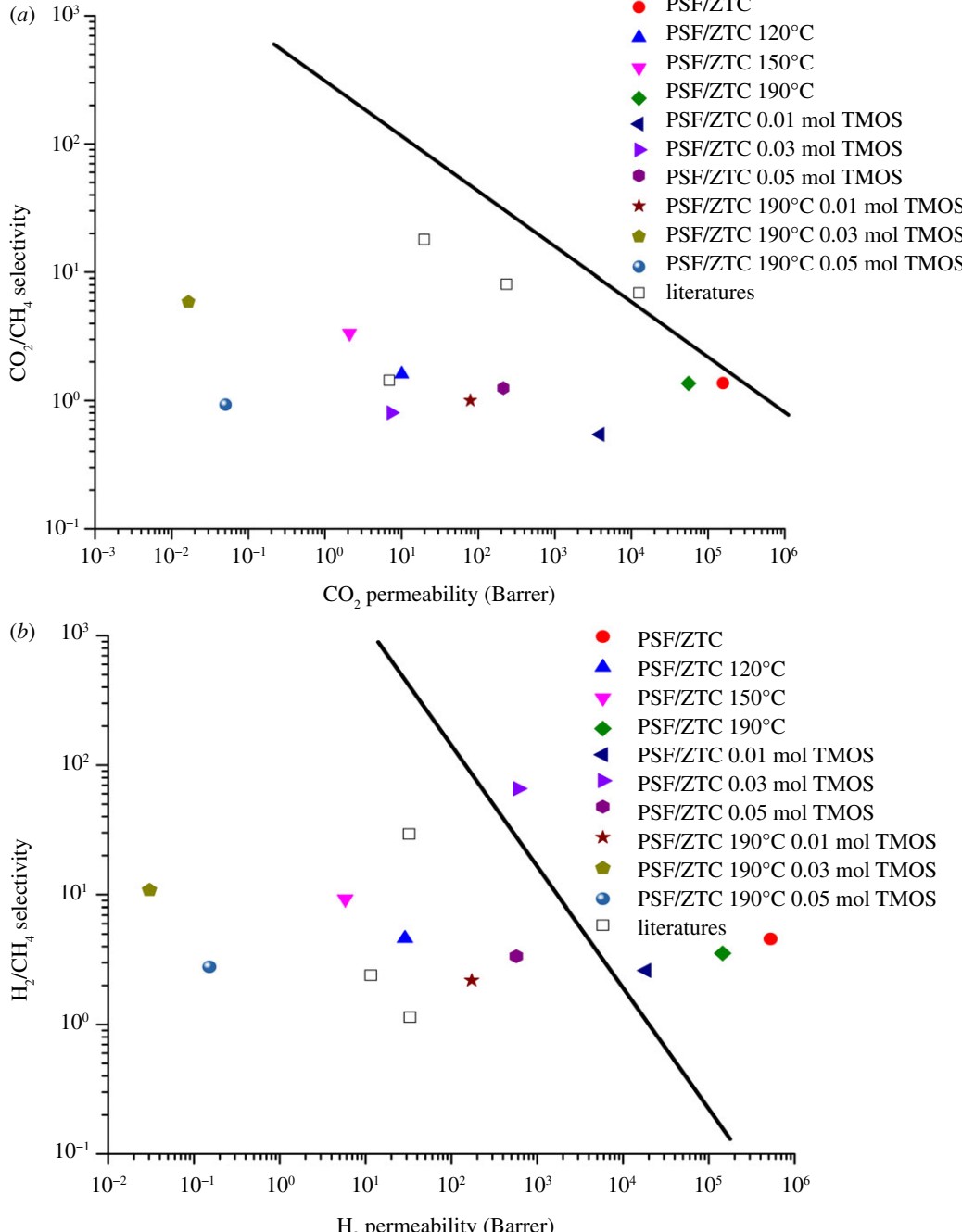

**Figure 11.** MMM PSF/ZTC gas separation performance with variations in heating temperature, coating concentration, and a combination of both for gas pairs (a) $CO_2/CH_4$ and (b) $H_2/CH_4$, compared to the Robeson curve [20] and other studies [19,67,68].

dissolving in the polymer matrix [63], while $CO_2$ and $CH_4$ gases tend to diffuse on the surface of the filler particles. Moreover, according to the SEM image, annealing reduces the size of the pore diameter in the polymer matrix. Hence, the gas with a large kinetic diameter ($CO_2$ and $CH_4$) is more difficult to diffuse across the membrane.

The diffusion mechanism that occurred on MMM PSF/ZTC coated with 0.01 mol TMOS was Knudsen diffusion because its ideal selectivity approached Knudsen's. Thus, the filler played an important role in the gas separation of MMM PSF/ZTC coated with 0.01 mol TMOS. The diffusion mechanism on MMM PSF/ZTC coated with 0.03 mol TMOS was a combination of surface diffusion and solution diffusion of the polymer due to the effect of gas affinity on the membrane matrix and the tighter structure, as shown by SEM analysis in figure 4g–i, as well as Knudsen diffusion of the filler as shown by its $CO_2/CH_4$ ideal selectivity, which was close to that of Knudsen. Knudsen also occurred on MMM PSF/ZTC coated with 0.05 mol TMOS.

The proposed diffusion mechanism on MMM PSF/ZTC annealed at 190°C and with a TMOS coating is a combination of Knudsen diffusion, surface flux, and solution diffusion. Different TMOS coating concentrations (0.01 and 0.05 mol) on annealed MMM exhibited low $CO_2/CH_4$ selectivity, namely 1.00 and 0.93, respectively. The low selectivity (similar to $CO_2/CH_4$ Knudsen selectivity of 0.6) indicates that the major gas diffusion contributor is Knudsen diffusion, in which the gas permeance is inversely related to the molecular weight of the penetrated species [64]. Furthermore, gas diffusion is influenced by the mean free path length, which is the average distance travelled by a gas molecule before colliding with another gas molecule [65]. Additionally, a similar trend was observed for $H_2/CH_4$ separation with 0.01 and 0.05 mol TMOS coating concentrations on annealed MMM, which means its selectivity is close to the $H_2/CH_4$ Knudsen selectivity (2.83). Hence, the Knudsen diffusion mechanism exerts a crucial influence on the gas separation mechanism. On the other hand, membrane selectivity beyond Knudsen selectivity was observed on an annealed MMM coated with 0.03 mol TMOS. Because the optimal silane coating on the membrane surface provides an electrical charge distribution on the membrane, it implies that the charge produces a difference in gas separation behaviour [66]. Takahashi *et al.* reported that at the surface of the porous alumina structure, oppositely charged atoms promote a physical interaction between the $CO_2$ molecule and the silane coupling agent [66]. This is to the fact that the $CH_4$ molecule is non-polar, while $CO_2$ has a polarity, thus providing a greater permeation of $CO_2$ than $CH_4$. Therefore, the other diffusion mechanism that makes a contribution is surface flux or facilitated diffusion.

A comparison between the performance of $CO_2/CH_4$ and $H_2/CH_4$ gas separation, and the Robeson upper-bound curve can be seen in figure 11 (electronic supplementary material, table S1). MMM PSF/ZTC without annealing and MMM PSF/ZTC annealed at 190°C had a gas separation performance of $CO_2/CH_4$ close to the Robeson upper-bound curve, and the $H_2/CH_4$ gas separation performance was above the Robeson upper-bound curve. These membranes exhibited a good separation performance, but the selectivity was still relatively low. On the other hand, MMM PSF/ZTC without coating and MMM PSF/ZTC coated with 0.01 and 0.03 mol TMOS had $H_2/CH_4$ gas separation performance above the Robeson upper limit, indicating good gas separation performance. This was unlike the $CO_2/CH_4$ gas separation performance, which was quite far from the Robeson upper limit. Furthermore, the gas separation performance of MMM PSF/ZTC modified with a combination of annealing and coating was no better than MMM PSF/ZTC modified by annealing or coating only. Therefore, the most appropriate membrane modification to improve the performance of MMM PSF/ZTC was annealing at 190°C or coating with 0.03 mol TMOS.

# 4. Conclusion

In this study, the gas separation performance of MMM PSF/ZTC was successfully improved. The annealing temperature and TMOS concentration affect membrane performance, whereby annealing reduces pore size as shown by SEM and XRD analysis. Coating with TMOS did not result in any chemical interaction between the membrane and TMOS, which was shown by the absence of changes to the FTIR spectra. MMM PSF/ZTC was modified by annealing at 120, 150, and 190°C; coating using 0.01, 0.03, and 0.05 mol TMOS; and a combination of both, with annealing at 190°C and coating using 0.03 mol TMOS. The $CO_2/CH_4$ selectivity of MMM PSF/ZTC was significantly improved, from 1.37 to 5.90 (331%), by a combination of annealing at 190°C and coating with 0.03 mol TMOS; similarly, $H_2/CH_4$ selectivity was improved, from 4.58 to 65.76 (1378%), by coating with 0.03 mol TMOS. The enhancement of selectivity was due to structural changes to the membrane that became denser and smoother, as observed by SEM and AFM. In this study, annealing and coating treatments are the best methods for improving the polymer matrix and filler particle adhesion. The $H_2/CH_4$ gas separation performances of MMM PSF/ZTC annealed at 190°C, and MMM PSF/ZTC coated with 0.01 and 0.03 mol TMOS were good because it was above the Robeson upper-limit curve. Permeation values for MMM PSF/ZTC annealed at 190°C were $H_2$: <H2: 107 433.20> 107 433.20 GPU, $CO_2$: 41 229.82 GPU, and $CH_4$: 30 293.33 GPU, with $H_2/CH_4$ and $CO_2/CH_4$ selectivity of 3.55 and 1.36, respectively. Meanwhile, the permeation values for MMM PSF/ZTC coated with 0.01 mol TMOS were $H_2$: 13 818.57 GPU, $CO_2$: 2887.87 GPU, and $CH_4$: 5297.68 GPU, with $H_2/CH_4$ and $CO_2/CH_4$ selectivity of 2.61 and 0.55, respectively. The permeation values for MMM PSF/ZTC coated with 0.03 mol TMOS were $H_2$: 444.11 GPU, $CO_2$: 5.42 GPU, and $CH_4$: 6.75 GPU, with $H_2/CH_4$ and $CO_2/CH_4$ selectivity of 65.76 and 0.8, respectively.

Ethical. This research did not involve human or animals for the object of study.

Data accessibility. Our dataset is deposited in the Dryad Digital Repository: https://doi.org/10.5061/dryad.m63xsj43s [69].
Authors' contributions. N.W.: conceptualization, supervision, writing—review and editing; I.S.C.: data curation, investigation, methodology, resources, writing—original draft; A.R.W.: investigation, resources, writing—review and editing; R.W.: conceptualization, methodology; T.G.: validation, writing—review and editing; Z.A.K.: formal analysis, resources; M.N.: supervision, writing—review and editing; Y.Y.: data curation, methodology, validation.

All authors gave final approval for publication and agreed to be held accountable for the work performed therein.
Conflict of interest declaration. We have no competing interests.
Funding. Kementerian Riset Teknologi dan Pendidikan Tinggi Indonesia research grant, contract no [135/SP2H/LT/DPRM/IV/2017]; and Penelitian Dasar by the Kementerian Pendidikan dan Kebudayaan Indonesia research grant, contract no [797/PKS/ITS/2021].
Acknowledgements. The authors would like to thank the *Kementerian Riset Teknologi dan Pendidikan Tinggi* Indonesia for PMDSU research funding, contract no [135/SP2H/LT/DPRM/IV/2017], and Penelitian Dasar by the *Kementerian Pendidikan dan Kebudayaan* Indonesia, contract no [1508/PKS/ITS/2022], for providing the research funding. The authors also would like to thank the *Pertamina* and *Pembangunan Perumahan* Company for providing the funding, and the Shibaura Institute of Technology for supporting Irmariza Shafitri Caralin with a student exchange scholarship.

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
