## [Peer Review File · Royal Society Open Science]

Review History

RSOS-211371.R0 (Original submission)

Review form: Reviewer 1

Is the manuscript scientifically sound in its present form?

No

Are the interpretations and conclusions justified by the results?

No

Is the language acceptable?

Yes

Do you have any ethical concerns with this paper?

No

Have you any concerns about statistical analyses in this paper?

No

Recommendation?

Reject

Comments to the Author(s)

this work reported that the development of ZTC MMMs for CO₂/CH₄ and H₂/CH₄ separations, the topic is very important for industrial application when it is applied to natural gas sweetening, and H₂ recovery from the natural gas grid. however, the prepared membranes do not provide a significant improvement compared to the state-of-the-art membranes reported in the literature or industrial applications. therefore, I feel this work did not provide significant scientific novelty for publishing in RSOS.

(1) based on the SEM images in Figure 4. it seems the ZTC particles caused a defect on the surface of the membranes. due to the large finger-like structure of the PSF MMMs, it is unlike the ZTC will provide any contribution neither to gas permeance nor selectivity. by post-treatment with annealing or coating TMOS selective to cause cross-link or make an extra selective layer to enhance the selective might provide some novelties, however, the significant reduction of gas permeance makes it unsuccessful for the post-treatment. as we can see from Figs. 7 and 8, those post-treated membranes do not present a better performance compared to the fresh membrane with respect to the Robeson upper bound. the slightly increased selectivity may not offset the significantly reduced permeance.

(2) the authors need to add the pure PSF membranes as the reference to document the effect of adding ZTC nanoparticles

(3) it is unclear why the membrane coated with 0.03 TMOS presented a much higher H₂/CH₄ selectivity compared to other membranes (≈ 10). this data point needs to be checked or at least explained. also, the CO₂/CH₄ selectivity is less than 1- unlikely (it might be the experimental error that there is no selectivity), error bar should be given.

(4) based on the SEM images of Fig. 1H and K, it is clear that the TMOS penetrates deeply into the matrix, which leads to the significant reduction of gas permeance, and of course a slight increase of the selectivity. The reviewer believes that the pore size of fresh PSF/ZTC MMMs is too large for direct coating.

Review form: Reviewer 2**Is the manuscript scientifically sound in its present form?**

Yes

Are the interpretations and conclusions justified by the results?

Yes

Is the language acceptable?

Yes

Do you have any ethical concerns with this paper?

No

Have you any concerns about statistical analyses in this paper?

No

Recommendation?

Accept with minor revision (please list in comments)

Comments to the Author(s)

The manuscript is more detailed, with clear and crisp tables of photographs. The materials and experimental methods are more complete and will facilitate research by other scholars. I consider it to be a manuscript that could be published.

Review form: Reviewer 3**Is the manuscript scientifically sound in its present form?**

Yes

Are the interpretations and conclusions justified by the results?

Yes

Is the language acceptable?

No

Do you have any ethical concerns with this paper?

No

Have you any concerns about statistical analyses in this paper?

No

Recommendation?

Major revision is needed (please make suggestions in comments)

Comments to the Author(s)

1. In Fig.1, the dense structure formed by the annealing treatment looks like a new layer formed on the PSF matrix, and H₂ CO₂ CH₄ were not marked out.
2. The manuscript lacks the characterization of ZTC itself, including but not limited to the pore size.
3. Whether it is possible to accurately characterize the pore size change range of the membrane before and after annealing.
4. The performance of binary gas H₂/CH₄ was only carried out using MMM annealed at 190°C, how about other types of membranes?
5. Please illustrate the reason of CH₄ increased permeance and CO₂ /CH₄ increased selectivity occurred on MMM coating with 0.05 mol TMOS.
6. "the excess TMOS concentration on the membrane was able to form multilayers, which could reduce adhesion between the polymer matrix and filler particles." The analysis is confused.
7. Paragraphs format are inconsistent and the serial number of CONCLUSION is error. Please check the whole manuscript carefully.
8. The logic of the manuscript does not match the order of figures placement and should be adjusted appropriately.
9. The preparation and application of MMM has not been reviewed completely, such as 10.1016/j.cej.2020.127144; 10.1016/j.seppur.2019.05.009; 10.1166/jnn.2017.13914; 10.1016/j.memsci.2016.07.060.
10. In Figure 4, the uppercase and lowercase letters are inconsistent with the legends, and there is no description for fig.4l.

Decision letter (RSOS-211371.R0)

Dear Dr Widiastuti:

Title: Selectivity Improvement of Polysulfone-Zeolite Templated Carbon Membrane by Annealing and Coating Treatment for CO₂/CH₄ and H₂/CH₄ Separation
Manuscript ID: RSOS-211371

The editor assigned to your manuscript has now received comments from reviewers. We would like you to revise your paper in accordance with the referee and Subject Editor suggestions which can be found below (not including confidential reports to the Editor). Please note this decision does not guarantee eventual acceptance.

Please submit your revised paper before 23-Feb-2022. Please note that the revision deadline will expire at 00.00am on this date. If we do not hear from you within this time then it will be assumed that the paper has been withdrawn. In exceptional circumstances, extensions may be possible if agreed with the Editorial Office in advance. We do not allow multiple rounds of revision so we urge you to make every effort to fully address all of the comments at this stage. If deemed necessary by the Editors, your manuscript will be sent back to one or more of the original reviewers for assessment. If the original reviewers are not available we may invite new reviewers.

Please also include the following statements alongside the other end statements. As we cannot publish your manuscript without these end statements included, if you feel that a given heading is not relevant to your paper, please nevertheless include the heading and explicitly state that it is not relevant to your work.

- Ethics statement

Please clarify whether you received ethical approval from a local ethics committee to carry out your study. If so please include details of this, including the name of the committee that gave consent in a Research Ethics section after your main text. Please also clarify whether you received informed consent for the participants to participate in the study and state this in your Research Ethics section.

OR

Please clarify whether you obtained the necessary licences and approvals from your institutional animal ethics committee before conducting your research. Please provide details of these licences and approvals in an Animal Ethics section after your main text.

OR

Please clarify whether you obtained the appropriate permissions and licences to conduct the fieldwork detailed in your study. Please provide details of these in your methods section.

- Data accessibility

It is a condition of publication that you make available the data and research materials supporting the results in the article. Datasets should be deposited in an appropriate publicly available repository and details of the associated accession number, link or DOI to the datasets must be included in the Data Accessibility section of the article

(<https://royalsocietypublishing.org/rsos/for-authors#question17>). Reference(s) to datasets should also be included in the reference list of the article with DOIs (where available).

Please include a Data Availability section after your main text stating where supporting data are available from, or where they will be made available should your article be accepted for publication.

<http://datadryad.org/submit?journalID=RSOS&manu=RSOS-211371>

- Competing interests

Please include a Competing Interests section after your main text declaring any financial or non-financial competing interests. If you have no competing interests please state 'I/we have no competing interests.'

- Authors' contributions

Please include an Authors' Contributions section at the end of your main text detailing the contribution of each author. All authors should have read and approved the manuscript before submission and this should be stated in the Authors' Contributions section.

The list of Authors should meet all of the following criteria; 1) substantial contributions to conception and design, or acquisition of data, or analysis and interpretation of data; 2) drafting the article or revising it critically for important intellectual content; and 3) final approval of the version to be published.

- Acknowledgements

- Funding statement

Please include a funding section after your main text which lists the source of funding for each author.

Yours sincerely,
Dr Ellis Wilde
Publishing Editor, Journals

On behalf of the Subject Editor Professor Anthony Stace and the Associate Editor Professor Chaohua Cui.

RSC Associate Editor
Comments to the Author:
(There are no comments.)

RSC Subject Editor
Comments to the Author:
(There are no comments.)

Reviewers' Comments to Author:

Reviewer: 1

Comments to the Author(s)

this work reported that the development of ZTC MMMs for CO₂/CH₄ and H₂/CH₄ separations, the topic is very important for industrial application when it is applied to natural gas sweetening, and H₂ recovery from the natural gas grid. however, the prepared membranes do not provide a significant improvement compared to the state-of-the-art membranes reported in the literature or industrial applications. therefore, I feel this work did not provide significant scientific novelty for publishing in RSOS.

(1) based on the SEM images in Figure 4. it seems the ZTC particles caused a defect on the surface of the membranes. due to the large finger-like structure of the PSF MMMs, it is unlike the ZTC will provide any contribution neither to gas permeance nor selectivity. by post-treatment with annealing or coating TMOS selective to cause cross-link or make an extra selective layer to enhance the selective might provide some novelties, however, the significant reduction of gas permeance makes it unsuccessful for the post-treatment. as we can see from Figs. 7 and 8, those post-treated membranes do not present a better performance compared to the fresh membrane with respect to the Robeson upper bound. the slightly increased selectivity may not offset the significantly reduced permeance.

(2) the authors need to add the pure PSF membranes as the reference to document the effect of adding ZTC nanoparticles

(3) it is unclear why the membrane coated with 0.03 TMOS presented a much higher H₂/CH₄ selectivity compared to other membranes (<10). this data point needs to be checked or at least explained. also, the CO₂/CH₄ selectivity is less than 1- unlikely (it might be the experimental error that there is no selectivity), error bar should be given.

(4) based on the SEM images of Fig. 1H and K, it is clear that the TMOS penetrates deeply into the matrix, which leads to the significant reduction of gas permeance, and of course a slight increase of the selectivity. The reviewer believes that the pore size of fresh PSF/ZTC MMMs is too large for direct coating.

Reviewer: 2

Comments to the Author(s)

The manuscript is more detailed, with clear and crisp tables of photographs. The materials and experimental methods are more complete and will facilitate research by other scholars. I consider it to be a manuscript that could be published.

Reviewer: 3

Comments to the Author(s)

1. In Fig.1, the dense structure formed by the annealing treatment looks like a new layer formed on the PSF matrix, and H₂ CO₂ CH₄ were not marked out.

2. The manuscript lacks the characterization of ZTC itself, including but not limited to the pore size.

3. Whether it is possible to accurately characterize the pore size change range of the membrane before and after annealing.

4. The performance of binary gas H₂/CH₄ was only carried out using MMM annealed at 190°C, how about other types of membranes?

5. Please illustrate the reason of CH₄ increased permeance and CO₂ /CH₄ increased selectivity occurred on MMM coating with 0.05 mol TMOS.

6. "the excess TMOS concentration on the membrane was able to form multilayers, which could reduce adhesion between the polymer matrix and filler particles." The analysis is confused.

7. Paragraphs format are inconsistent and the serial number of CONCLUSION is error. Please check the whole manuscript carefully.

8. The logic of the manuscript does not match the order of figures placement and should be adjusted appropriately.

9. The preparation and application of MMM has not been reviewed completely, such as 10.1016/j.cej.2020.127144; 10.1016/j.seppur.2019.05.009; 10.1166/jnn.2017.13914; 10.1016/j.memsci.2016.07.060.

10. In Figure 4, the uppercase and lowercase letters are inconsistent with the legends, and there is no description for fig.4l.

Author's Response to Decision Letter for (RSOS-211371.R0)

See Appendix A.

RSOS-211371.R1 (Revision)

Review form: Reviewer 1

Is the manuscript scientifically sound in its present form?

Yes

Are the interpretations and conclusions justified by the results?

Yes

Is the language acceptable?

Yes

Do you have any ethical concerns with this paper?

No

Have you any concerns about statistical analyses in this paper?

No

Recommendation?

Accept as is

Comments to the Author(s)

1) the title might be a bit confusing as the reader might confuse with carbon membranes (in fact this is a PSF/ZTC MMM)
2) the title indicated the enhancement for CO₂/CH₄ separation by annealing and coating treatment. however, from Table 5, the reviewer cannot clearly see the improvement at least for CO₂/CH₄ separation, for some cases, it decreases. and the authors may not fully address my comment why CO₂/CH₄ selectivity is lower than 1. what is the transport mechanism.

Review form: Reviewer 2

Is the manuscript scientifically sound in its present form?

Yes

Are the interpretations and conclusions justified by the results?

Yes

Is the language acceptable?

Yes

Do you have any ethical concerns with this paper?

No

Have you any concerns about statistical analyses in this paper?

No

Recommendation?

Accept as is

Comments to the Author(s)

The author is familiar with is manuscript content and revises it carefully and completely with a serious attitude. The structure of the article is clear. Recommended for acceptance.

Decision letter (RSOS-211371.R1)

Dear Dr Widiastuti:

Title: Selectivity Improvement of Polysulfone-Zeolite Templated Carbon Membrane by Annealing and Coating Treatment for CO₂/CH₄ and H₂/CH₄ Separation
Manuscript ID: RSOS-211371.R1

Thank you for submitting the above manuscript to Royal Society Open Science. On behalf of the Editors and the Royal Society of Chemistry, I am pleased to inform you that your manuscript will be accepted for publication in Royal Society Open Science subject to minor revision in accordance with the referee suggestions. Please find the reviewers' comments at the end of this email.

The reviewers and handling editors have recommended publication, but also suggest some minor revisions to your manuscript. Therefore, I invite you to respond to the comments and revise your manuscript.

Please also include the following statements alongside the other end statements. As we cannot publish your manuscript without these end statements included, if you feel that a given heading is not relevant to your paper, please nevertheless include the heading and explicitly state that it is not relevant to your work. We have included a screenshot example of the end statements for reference.

- Ethics statement

Please clarify whether you received ethical approval from a local ethics committee to carry out your study. If so please include details of this, including the name of the committee that gave consent in a Research Ethics section after your main text. Please also clarify whether you received informed consent for the participants to participate in the study and state this in your Research Ethics section.

OR

Please clarify whether you obtained the necessary licences and approvals from your institutional animal ethics committee before conducting your research. Please provide details of these licences and approvals in an Animal Ethics section after your main text.

OR

Please clarify whether you obtained the appropriate permissions and licences to conduct the fieldwork detailed in your study. Please provide details of these in your methods section.

- Data accessibility

It is a condition of publication that you make available the data and research materials supporting the results in the article. Datasets should be deposited in an appropriate publicly available repository and details of the associated accession number, link or DOI to the datasets must be included in the Data Accessibility section of the article (<https://royalsocietypublishing.org/rsos/for-authors#question17>). Reference(s) to datasets should also be included in the reference list of the article with DOIs (where available).

Please include a Data Availability section after your main text stating where supporting data are available from, or where they will be made available should your article be accepted for publication.

If you wish to submit your supporting data or code to Dryad (<http://datadryad.org/>), or modify your current submission to dryad, please use the following link:
<http://datadryad.org/submit?journalID=RSOS&manu=RSOS-211371.R1>

- **Competing interests**

Please include a Competing Interests section after your main text declaring any financial or non-financial competing interests. If you have no competing interests please state 'I/we have no competing interests.'

- **Authors' contributions**

Please include an Authors' Contributions section at the end of your main text detailing the contribution of each author. All authors should have read and approved the manuscript before submission and this should be stated in the Authors' Contributions section.

The list of Authors should meet all of the following criteria; 1) substantial contributions to conception and design, or acquisition of data, or analysis and interpretation of data; 2) drafting the article or revising it critically for important intellectual content; and 3) final approval of the version to be published.

- **Acknowledgements**

- **Funding statement**

Please include a funding section after your main text which lists the source of funding for each author.

Because the schedule for publication is very tight, it is a condition of publication that you submit the revised version of your manuscript before 08-May-2022. Please note that the revision deadline will expire at 00.00am on this date. If you do not think you will be able to meet this date please let me know immediately.

Kind regards,
Ellis Wilde and Kate Jones
Assistant Editor, Journals

On behalf of the Subject Editor Professor Anthony Stace and the Associate Editor Professor Chaohua Cui.

RSC Associate Editor: 1
Comments to the Author:
(There are no comments.)

RSC Associate Editor: 2
Comments to the Author:
(There are no comments.)

Reviewer comments to Author:

Reviewer: 1

Comments to the Author(s)

- 1) the title might be a bit confusing as the reader might confuse with carbon membranes (in fact this is a PSF/ZTC MMM)
- 2) the title indicated the enhancement for CO₂/CH₄ separation by annealing and coating treatment. however, from Table 5, the reviewer cannot clearly see the improvement at least for CO₂/CH₄ separation, for some cases, it decreases. and the authors may not fully address my comment why CO₂/CH₄ selectivity is lower than 1. what is the transport mechanism.

Reviewer: 2

Comments to the Author(s)

The author is familiar with is manuscript content and revises it carefully and completely with a serious attitude. The structure of the article is clear. Recommended for acceptance.

Author's Response to Decision Letter for (RSOS-211371.R1)

See Appendix B.

RSOS-211371.R2

Review form: Reviewer 1

Is the manuscript scientifically sound in its present form?

Yes

Are the interpretations and conclusions justified by the results?

Yes

Is the language acceptable?

Yes

Do you have any ethical concerns with this paper?

No

Have you any concerns about statistical analyses in this paper?

No

Recommendation?

Accept as is

Comments to the Author(s)

I'm satisfied with the revision, and recommend to publishing this work.

Decision letter (RSOS-211371.R2)

Dear Dr Widiastuti:

Title: Annealing and TMOS coating on PSF/ZTC mixed matrix membrane for enhanced CO₂/CH₄ and H₂/CH₄ separation
Manuscript ID: RSOS-211371.R2

It is a pleasure to accept your manuscript in its current form for publication in Royal Society Open Science. The chemistry content of Royal Society Open Science is published in collaboration with the Royal Society of Chemistry.

Where applicable, the comments of the reviewer(s) who reviewed your manuscript are included at the end of this email.

If you have not already done so, please ensure that you send to the editorial office (openscience@royalsociety.org) an editable version of your accepted manuscript, and individual files for each figure and table included in your manuscript. You can send these in a zip folder if more convenient. Failure to provide these files may delay the processing of your proof.

Please remember to make any data sets or code libraries 'live' prior to publication, and update any links as needed when you receive a proof to check - for instance, from a private 'for review' URL to a publicly accessible 'for publication' URL. It is also good practice to add data sets, code and other digital materials to your reference list.

Royal Society Open Science is a fully open access journal. A payment may be due before your article is published. Our partner Copyright Clearance Centre will contact the corresponding author about your open access options (if you have any queries regarding fees, please see <https://royalsocietypublishing.org/rsos/charges> or contact authorfees@royalsociety.org).

Yours sincerely,
Raffaele Egizio

Assistant Editor, Journals

On behalf of the Subject Editor Professor Anthony Stace and the Associate Editor Professor
Chaohua Cui.

RSC Associate Editor
Comments to the Author:
(There are no comments.)

RSC Subject Editor
Comments to the Author:
(There are no comments.)

Reviewer(s)' Comments to Author:
Reviewer: 1
Comments to the Author(s)
I'm satisfied with the revision, and recommend to publishing this work.

Appendix A

RE: Point-by-point response for revision to manuscript RSOS-211371

Title: Selectivity Improvement of Polysulfone-Zeolite Templated Carbon Membrane by Annealing and Coating Treatment for CO₂/CH₄ and H₂/CH₄ Separation

Authors: Nurul Widiastuti, Irmaliza Shafitri Caralin, Alvin Rahmad Widyanto, Rika

Wijiyanti, Triyanda Gunawan, Zulhairun Abdul Karim, Mikihiro Nomura, Yuki Yoshida

February 3rd, 2022

Response to Reviewers' comments:

We would like to thank the Editor and the Reviewers for their constructive comments and suggestions to help us improve our research and the quality of this manuscript. We have carefully considered all the Reviewer's comments and have revised the manuscript to address their concerns. To aid in the reviewing process, we have replied to all the comments on a point-by-point basis and highlighted the revised sections of the main manuscript with blue font color. We hope the manuscript can now be accepted for publishing in Royal Society Open Science.

On behalf of the authors, and with kind regards,

Dr. Nurul Widiastuti (corresponding author for the submission)

Response to Reviewers of Royal Society Open Science

	Reviewer #1: this work reported that the development of ZTC MMMs for CO ₂ /CH ₄ and H ₂ /CH ₄ separations, the topic is very important for industrial application when it is applied to natural gas sweetening, and H ₂ recovery from the natural gas grid. however, the prepared membranes do not provide a significant improvement compared to the state-of-the-art membranes reported in the literature or industrial applications. therefore, I feel this work did not provide significant scientific novelty for publishing in RSOS.
1	based on the SEM images in Figure 4. it seems the ZTC particles caused a defect on the surface of the membranes. due to the large finger-like structure of the PSF MMMs, it is unlikely the ZTC will provide any contribution neither to gas permeance nor selectivity. by post-treatment with annealing or coating TMOS selective to cause cross-link or make an extra selective layer to enhance the selective might provide some novelties, however, the significant reduction of gas permeance makes it unsuccessful for the posttreatment. as we can see from Figs. 7 and 8, those post-treated membranes do not present a better performance compared to the fresh membrane with respect to the Robeson upper bound. the slightly increased selectivity may not offset the significantly reduced permeance.
	Response First of all, thank you for your time to review this manuscript. We greatly appreciate the reviewer's positive comments. Our response to your comments is below: There is no relationship between the defects and the finger-like structure. Based on Wijiyanti et al. [1], the neat membrane does have finger-like pores similar to this study. The finger-like pore was obtained as a result of phase inversion between coagulation

liquid and polymer solution during the dry/wet-spinning process. Thus, it cannot be expressed as a defect. The defect that occurs in MMM is in the form of interfacial voids, which is shown in Figure b. However, in this study, there is no visible defect on the membrane, as shown in Figures 4b1, d1, f1.

Page 5 Line 36-39.

According to the SEM observation, the finger-like pore was formed during the dry/wet-spinning process as a consequence of phase inversion between the coagulation liquid and polymer solution. On the other hand, the presence of voids in MMM PSF/ZTC without annealing, as shown in Wijiyanti et al. [1], was due to the low adhesion between the polymer matrix and the ZTC.

Figure. SEM Image of (a) neat PSF membrane (cross-section), (b) unmodified MMM PSF/ZTC (surface) from [1]; 150 °C annealed MMM PSF/ZTC (c) cross-section, and (d) surface in this study.

Indeed, the annealing and TMOS coating process does not provide a significant increment in gas separation performance. However, it should be noted that annealing is important in improving the mechanical properties of membranes by promoting the strengthened interactions among polymer chains, and a higher degree of crystallinity in the polymer matrix. Pham et al. [2] reported annealing at 100, 150, and 200 °C can increase the mechanical strength of membrane from 372 to 586, 734, and 743 MPa, respectively.

	Page 7 Line 20-23. On the other hand, annealing improved the mechanical characteristics of polymeric membranes by encouraging stronger connections between polymer chains and a greater degree of crystallinity in the polymer matrix. Annealing the membrane at 100, 150, or 200°C enhanced its mechanical strength from 372 to 586, 734, or 743 MPa, respectively [2]. On the other hand, the addition of TMOS did cause a decrease in permeability, which can be shown by SEM images which are much denser in structure (Figure 4g, h). This led to a significant increase in selectivity in the membrane with 0.03 mol TMOS for the studied gas separation. Furthermore, a significant increase in selectivity occurred in the gas separation process with different gas sizes, especially in the H₂/CH₄ gas separation. As a result, the addition of TMOS in an optimum composition could increase membrane performance. For instance, the membrane coated with 0.03 mol TMOS exhibits superior H₂/CH₄ separation performance.
2	the authors need to add the pure PSF membranes as the reference to document the effect of adding ZTC nanoparticles.
	Response We highly recognize the reviewer's positive comments. However, in this study, we focus on the investigation of post-treatment MMM PSF/ZTC to improve gas separation performance. We have been discussed the effect of adding ZTC fillers in the PSF membrane, as well as the neat PSF membrane characterization that has been exhibited in the previous studies [1,3]. Furthermore, we have discussed the addition ZTC nanoparticle could enhanced CO₂/CH₄ and H₂/CH₄ gasses separation performance of polysulfone membrane. Page 2 Line 39-41. Our result showed that the presence of ZTC as filler in the MMM-based polysulfone increased the selectivity of CO₂/CH₄ from 2.56 to 9.99 and selectivity of H₂/CH₄ from 7.77 to 28.88 [1].
3	it is unclear why the membrane coated with 0.03 TMOS presented a much higher H₂/CH₄ selectivity compared to other membranes (<10). this data point needs to be checked or at least explained. also, the CO₂/CH₄ selectivity is less than 1- unlikely (it might be the experimental error that there is no selectivity), error bar should be given.

Response

We highly recognize the reviewer's positive comments, we have added error bar in all Figures and standard deviation in Table 2.

Page 16.

Table 2. Permeation and selectivity of single gases on MMM PSF/ZTC with variations of annealing temperature, coating treatment and its combination.

Membranes	Permeation (GPU)			Selectivity	
	H ₂	CO ₂	CH ₄	H ₂ /CH ₄	CO ₂ /C H ₄
MMM PSF/ZTC	389775.2 9 ±	116361.0 3 ±	85088.9 1 ±	4.58	1.37
	46321.49	3477.65	3392.87		
MMM PSF/ZTC annealed at 120°C	4.69 ±	1.63 ±	1.01 ±	4.63 (0.97%)	1.61 (17%)
	0.041	0.007	0.013		
	(-99%)	(-99%)	(-99%)		
MMM PSF/ZTC annealed at 150°C	3.89 ±	1.41 ±	0.42 ±	9.26 (102%)	3.35 (144%)
	0.054	0.003	0.001		
	(-99%)	(-99%)	(-99%)		
MMM PSF/ZTC annealed at 190°C	107433.2 0 ±	41229.82 ±	30293.3 3 ±	3.55 (-22%)	1.36 (- 0.48%)
	13317.84	1683.41	1033.68		
	(-72%)	(-64%)	(-64%)		
MMM PSF/ZTC coated with 0.01 mol TMOS	13818.57 ±	2887.87 ±	5297.68 ±	2.61 (-43%)	0.55 (-60%)
	420.95	36.64	58.03		
	(-96%)	(-97%)	(-93%)		
MMM PSF/ZTC coated with 0.03 mol TMOS	444.11 ±	5.42 ±	6.75 ±	65.76 (1335.52%)	0.80 (-41%)
	7.76	0.10	0.10		
	(-99%)	(-99%)	(-99%)		
MMM PSF/ZTC coated with 0.05 mol TMOS	423.67 ±	157.35 ±	125.88 ±	3.37 (-26%)	1.25 (-8%)
	17.83	2.27	2.57		
	(-99%)	(-99%)			

				(-99%)	
MMM PSF/ZTC 190°C	128.08 ±	58.55 ±	58.47 ±	2.19	1.00
0.01 mol TMOS	1.80	4.39	1.70	(-52%)	(-26%)
	(-99%)	(-99%)	(-99%)		
MMM PSF/ZTC 190°C	0.02 ±	0.01 ±	0.002 ±	10.83	5.90
0.03 mol TMOS	0.0002	0.0001	0.0001	(136%)	(331%)
	(-99%)	(-99%)	(-100%)		
MMM PSF/ZTC 190°C	0.11 ±	0.037 ±	0.040 ±	2.79	0.93
0.05 mol TMOS	0.0006	0.0002	0.0007	(-39%)	(-32%)
	(-99%)	(-99%)	(-100%)		
Knudsen selectivity				2.83	0.6

We suggest that 0.03 mol TMOS addition is provided an ideal coating for the membrane to separate H₂/CH₄. Furthermore, for CO₂/CH₄ gas separation, the addition of 0.05 TMOS gives the best results in terms of selectivity, but if we look at the trend, it is actually still necessary to carry out further variations on the higher addition of TMOS concentration (which will be considered in future research). According to the H₂/CH₄ separation performance, the addition of TMOS is optimal at 0.03 mol. We guess the increase in concentration is the optimal condition to produce a membrane with the appropriate structure in the H₂/CH₄ separation process. It is also exhibit similar result from Kagari et al. [4], in which they study coating PEI membrane utilizes PDMS at various concentrations (5-17 wt%). The selectivity of H₂/CH₄ was increased utilizing coating solutions up to 15.0 wt%, but thereafter decreased as the concentration increased. Further increasing the coating solution concentration seems to have resulted in a thicker layer with low gas permeance. On the other hand, the selection size of the silane coating agent influences the gas diffusion compatibility. For example, PVP was not suited for surface modification of microporous inorganic fillers since polymer sizing on the particle surface is prone to causing pore blockage [5]. This argument is used for 0.03 mol TMOS utilization, which could be disturbing the diffusion of higher size of gas molecules (i.e., CO₂ and CH₄) due to the membrane pores are getting narrower, while H₂ penetration not affected considerably.

We have added discussion.

Page 3 Line 9-13.

On the other hand, the selection size of the silane coating agent influences the gas diffusion compatibility. For example, PVP was not suited for surface modification of microporous inorganic fillers since polymer sizing on the particle surface is prone to

	causing pore blockage [5]. Thus, this study discovered the other potential coating material which has unique properties to enhance gas separation performance. Page 8 Line 34-36. It suggests that at 0.03 mol TMOS utilization could be disturbing the diffusion of higher size of gas molecules (i.e., CO₂ and CH₄) due to the membrane pores are getting narrower, while H₂ penetration not affected considerably. As seen in Figure 7 (a-c), the MMM pore was smaller.
4	based on the SEM images of Fig. 1H and K, it is clear that the TMOS penetrates deeply into the matrix, which leads to the significant reduction of gas permeance, and of course a slight increase of the selectivity. The reviewer believes that the pore size of fresh PSF/ZTC MMMs is too large for direct coating.
	Response We are aware of it and we also would like to suggest for coating should only be dyed. We do it under reflux expecting a cross link between the TMOS and the membrane. However, we did not find the cross-linking behavior according to the FTIR spectra.

Reviewer #2: The manuscript is more detailed, with clear and crisp tables of photographs. The materials and experimental methods are more complete and will facilitate research by other scholars. I consider it to be a manuscript that could be published.	
	Response First of all, thank you for your time to review this manuscript. Thank you for your positive feedback. We highly appreciate it.

Reviewer #3:	
1	In Fig.1, the dense structure formed by the annealing treatment looks like a new layer formed on the PSF matrix, and H₂ CO₂ CH₄ were not marked out.
	Response First of all, thank you for your time to review this manuscript. The reviewer raises an interesting concern. Thank you very much for your positive comment, we have been redrawing the figure. Sorry for making it confusing, but it's not. Please kindly see the revised illustration for a better understanding.

Figure. Revised illustration which exhibited in Fig 1.

2	The manuscript lacks the characterization of ZTC itself, including but not limited to the pore size.
	Response We highly recognize the reviewer’s positive comments. However, we have been discussed more detailed ZTC characterization in previous research in [1,3,6].
3	Whether it is possible to accurately characterize the pore size change range of the membrane before and after annealing.
	Response We highly recognize the reviewer’s positive comments. Tsuru and coworkers have been reported the utilization of nanoporometry apparatus to determine the membrane pore size [7–11]. However, we did not have access to conducting this characterization currently. We would consider studying the pore size change analyzation of the mixed matrix membrane in further research as it is also the important feature for aid elucidating the separation processes in membrane.
4	The performance of binary gas H2/CH4 was only carried out using MMM annealed at 190°C, how about other types of membranes?
	Response We highly recognize the reviewer’s positive comments. We chose a membrane that has been annealed at 190 °C since it has the highest separation performance in single gas separation, and we presume that mixed gas separation is not much different.
5	Please illustrate the reason of CH4 increased permeance and CO2 /CH4 increased selectivity occurred on MMM coating with 0.05 mol TMOS.
	Response We apologize with the data drawn on the graph is upside down for CO2 and CH4 permeation (Figure 6a). Thus, we have redrawn it and rechecked all data carefully.

Figure. 6 (a) Permeation and selectivity of (a) CO₂/CH₄ (Page 9 Line 1)

As the concentration of the TMOS coating increases, it covers the membrane pores so that the most influential diffusion is solution diffusion, which is more influenced by the solubility of the gas. Therefore, the H₂ gas permeation decreased while the CO₂ and CH₄ gas permeation increased. Furthermore, we suggest that the addition of silane which consist of oxygen atom promotes physical interaction due to higher polarity [12,13]. Thus, CO₂ permeation higher than CH₄ as well as the CO₂/CH₄ selectivity increase following by increasing TMOS concentration coating.

We have added this discussion below, Page 8 Line 44-50:

However, at 0.05 mol TMOS coating it observed a unique pattern that the penetration of H₂ gas reduced while the permeability of CO₂ and CH₄ gas climbed. Because this concentration of the TMOS coating rises, it completely covers the membrane pores, favoring solution diffusion, which is more dependent on the gas's solubility. Moreover, it suggests that the incorporation of silane, which contains oxygen atoms, facilitates physical contact owing to its increased polarity [12,13]. Thus, increasing the TMOS concentration coating leads to a rise in CO₂ permeation greater than CH₄ permeation, as a result, CO₂/CH₄ selectivity improved

6 “the excess TMOS concentration on the membrane was able to form multilayers, which could reduce adhesion between the polymer matrix and filler particles.” The analysis is confused.

Response

We highly recognize the reviewer’s positive comments. Thank you for your concern. We have revised the sentences.

Page 8 Line 40-41.

	the excess TMOS concentration on the membrane was able to form multilayers, which can not only cover the voids but also block the gas diffusion path. Page 8 Line 43-44. Furthermore, higher Pebax coating concentration (9wt%) on PSF membrane decreased CO₂ permeation to 11.55 GPU from 47.73 GPU (1wt%) [14].
7	Paragraphs format are inconsistent and the serial number of CONCLUSION is error. Please check the whole manuscript carefully.
	Response Thank you for your feedback, we have revised it, Page 12 Line 13. 5. Conclusion
8	The logic of the manuscript does not match the order of figures placement and should be adjusted appropriately.
	Response We highly recognize the reviewer's positive comments, our response to your comments below: Thank you for the constructive suggestion, we have reorganized it. From Figure 4. Figure 4. SEM morphology of MMM PSF/KTZ's cross-section annealed at (a) 120 °C, (c) 150 °C, (e) 190 °C, and its surfaces annealed at (b) 120 °C; and (b1) zoom-in of (b), (d) 150 °C; and (d1) zoom-in of (d), (f) 190 °C; and (f1) zoom-in of (f), with magnification of 850-33000x; coated TMOS (g, h) cross-section and its surface (i) with magnification of 180-2000x; annealed at 190 °C and coated TMOS (j, k) cross section, and its surface (e) with magnification of 230-2600x. Removed the figure 4(g-k) to other section, becoming: Page 6 Line 19-20.

Figure 4. SEM morphology of MMM PSF/KTZ's cross-section annealed at (a) 120 °C, (c) 150 °C, (e) 190 °C, and its surfaces annealed at (b) 120 °C; and (b1) zoom-in of (b), (d) 150 °C; and (d1) zoom-in of (d), (f) 190 °C; and (f1) zoom-in of (f).

In addition, Figure 5 create from combining of XRD, TGA and AFM characterization of annealing membrane.

Page 7 Line 32.

Figure 5. MMM PSF/ZTC characterizations of (a) XRD diffractogram ((a) without annealing, annealed at (b) 120 °C, (c) 150 °C, and (d) 190 °C); (b) TGA curve; AFM Images of (c) without modification; annealed at (d) 120 °C, (e)150 °C, and (f)190 °C.

Furthermore, Figure 7 create from combining of SEM, AFM and FTIR characterization of coating membrane.

Page 9 Line 5.

Figure 7. Coated MMM PSF/ZTC characterization of (a,b) cross-section, and (c) its surface SEM Images; (d) AFM Images; and (e) FTIR spectra.

Additionally, Figure 9 create from combining of SEM, and AFM characterization of annealing and coating combination post-treatment in membrane.

Page 10 Line 24.

Figure 9. Double post-treatment (annealing and coating) on MMM PSF/ZTC characterization of (a,b) cross-section and (c) its surface SEM images; and (d) AFM images.

Lastly, the Robeson upper bound discussion move to Page

The comparison between the performance of CO₂/CH₄ and H₂/CH₄ gas separation, and the Robeson upper bound curve can be seen in Figure 11. MMM PSF/ZTC without annealing and MMM PSF/ZTC annealed at 190 °C had a gas separation performance of CO₂/CH₄ close to the Robeson upper bound curve and H₂/CH₄ gas separation performance, which was above the Robeson upper bound curve. Those exhibited a good separation performance, but the selectivity was still relatively low. On the other hand, The MMM PSF/ZTC without coating and MMM PSF/ZTC coated with 0.01 and 0.03 mol TMOS had H₂/CH₄ gas separation performance was above the Robeson upper limit, indicating good gas separation performance. Unlike the case with the CO₂/CH₄ gas separation performance, which was quite far from the Robeson upper limit. Furthermore, the gas separation performance of MMM PSF/ZTC modified with combinations of annealing and coating was no better than MMM PSF/ZTC modified by annealing or coating only. Therefore, the appropriate membrane modification to improve the performance of MMM PSF/ZTC was annealing at 190 °C or coating with 0.03 mol TMOS.

Figure 11. MMM PSF/ZTC gas separation performance with variations in heating temperature, coating concentration, and its combination for gas pairs (a) CO₂/CH₄ and (b) H₂/CH₄, compared to the Robeson curve [15] and other studies [16–18].

9	The preparation and application of MMM has not been reviewed completely, such as 10.1016/j.cej.2020.127144; 10.1016/j.seppur.2019.05.009; 10.1166/jnn.2017.13914; 10.1016/j.memsci.2016.07.060.
	Response We highly recognize the reviewer's positive comments. However, the cited paper is not relevant with this study. In this study we focused on mixed matrix membrane PSF/ZTC for gas separation application. The study from (10.1016/j.cej.2020.127144; 10.1016/j.seppur.2019.05.009; 10.1166/jnn.2017.13914; 10.1016/j.memsci.2016.07.060) was focus on nanofiltration and ultrafiltration. Therefore, we regret that we could not cite it since the topics discussed are wide and do not focus on gas separation applications. Furthermore, we have been reviewed the previous study of MMM for gas separation. Our result showed that the presence of ZTC as filler in the MMM-based polysulfone increased the selectivity of CO₂/CH₄ from 2.56 to 9.99 and selectivity of H₂/CH₄ from 7.77 to 28.88 [1]. The other fillers applied in MMM based polysulfone have been reported by several researchers such as zeolite and silica. Mohamat et al. [19] reported that the incorporation of 3wt% zeolite T in polysulfone membrane enhanced CO₂/CH₄ selectivity from 2.63 to 3.37 with CO₂ permeability as 78.90 GPU. On the other hand, the presence 2wt% of KIT-6 (KIT: Korea Advanced Institute of Science and Technology), a silica mesoporous, could improve CO₂/CH₄ selectivity for 32.4 with CO₂ permeability as 5.4 Barrer [20]. On the other hand, thought the preparation method of this study is utilizing phase inversion, it's quite different preparation technique in this study and the cited literature. Since in this study was used hollow fiber module and utilizing different solvent and polymer matrix, the cited literatures (10.1016/j.cej.2020.127144; 10.1016/j.seppur.2019.05.009; 10.1166/jnn.2017.13914; 10.1016/j.memsci.2016.07.060) used DMAc as a solvent to produce flat sheet mixed matrix membranes based PVDF or PES polymer matrix. In addition, we have been cited more similar preparation of hollow fiber MMM that adopted from previous studies. Page 4 Line 36-37. The membrane preparation is adopted from the previous literatures [1,3].
10	In Figure 4, the uppercase and lowercase letters are inconsistent with the legends, and there is no description for fig.4l.
	Response

Thank you for your concern about it, we have revised it. Please kindly see revised Figure 4.

Page 6 Line 19.

Furthermore, we have added legend for zoom in image of (b,d, and f).

Page 6 Line 20-22.

Figure 4. SEM morphology of MMM PSF/KTZ's cross-section annealed at (a) 120 °C, (c) 150 °C, (e) 190 °C, and its surfaces annealed at (b) 120 °C; and (b1) zoom-in of (b), (d) 150 °C; and (d1) zoom-in of (d), (f) 190 °C; and (f1) zoom-in of (f).

REFERENCES

- [1] R. Wijiyanti, A.N. Ubaidillah, T. Gunawan, Z.A. Karim, A.F. Ismail, S. Smart, R. Lin, N. Widiastuti, Polysulfone mixed matrix hollow fiber membranes using zeolite templated carbon as a performance enhancement filler for gas separation, *Chem. Eng. Res. Des.* 150 (2019) 274–288. <https://doi.org/10.1016/j.cherd.2019.08.004>.
- [2] T.A. Pham, S. Koo, H. Park, Q.T. Luong, O.J. Kwon, S. Jang, S.M. Kim, K. Kim, Investigation on the Microscopic/Macroscopic Mechanical Properties of a Thermally Annealed Nafion® Membrane, *Polymers (Basel)*. 13 (2021) 4018. <https://doi.org/10.3390/polym13224018>.
- [3] R. Wijiyanti, A.R. Kumala Wardhani, R.A. Roslan, T. Gunawan, Z. Abdul Karim, A.F. Ismail, N. Widiastuti, Enhanced gas separation performance of polysulfone membrane by incorporation of zeolite-templated carbon, *Malaysian J. Fundam. Appl. Sci.* 16 (2020) 128–134. <https://doi.org/10.11113/mjfas.v16n2.1472>.
- [4] A. Kargari, A. Arabi Shamsabadi, M. Bahrami Babaheidari, Influence of coating conditions on the H₂ separation performance from H₂/CH₄ gas mixtures by the PDMS/PEI composite membrane, *Int. J. Hydrogen Energy*. 39 (2014) 6588–6597. <https://doi.org/10.1016/j.ijhydene.2014.02.009>.
- [5] N.N. Rosyadah Ahmad, H. Mukhtar, D.F. Mohshim, R. Nasir, Z. Man, Surface modification in inorganic filler of mixed matrix membrane for enhancing the gas

- separation performance, *Rev. Chem. Eng.* 32 (2016) 181–200. <https://doi.org/10.1515/revce-2015-0031>.
- [6] T. Gunawan, R. Wijiyanti, N. Widiastuti, Adsorption-desorption of CO₂ on zeolite-Y-templated carbon at various temperatures, *RSC Adv.* 8 (2018) 41594–41602. <https://doi.org/10.1039/c8ra09200a>.
- [7] J. Wang, Y. Ma, T. Tsuru, Prediction of pervaporation performance of aqueous ethanol solutions by nanoporometry characterization, *Sep. Purif. Technol.* 74 (2010) 310–317. <https://doi.org/10.1016/j.seppur.2010.06.021>.
- [8] X. Ren, M. Kanezashi, H. Nagasawa, T. Tsuru, Preparation of organosilica membranes on hydrophobic intermediate layers and evaluation of gas permeation in the presence of water vapor, *J. Memb. Sci.* 496 (2015) 156–164. <https://doi.org/10.1016/j.memsci.2015.08.050>.
- [9] T. Tsuru, T. Hino, T. Yoshioka, M. Asaeda, Porometry characterization of microporous ceramic membranes, *J. Memb. Sci.* 186 (2001) 257–265. [https://doi.org/10.1016/S0376-7388\(00\)00692-X](https://doi.org/10.1016/S0376-7388(00)00692-X).
- [10] T. Tsuru, Y. Takata, H. Kondo, F. Hirano, T. Yoshioka, M. Asaeda, Characterization of sol-gel derived membranes and zeolite membranes by nanoporometry, *Sep. Purif. Technol.* 32 (2003) 23–27. [https://doi.org/https://doi.org/10.1016/S1383-5866\(03\)00036-4](https://doi.org/https://doi.org/10.1016/S1383-5866(03)00036-4).
- [11] N. Moriyama, H. Nagasawa, M. Kanezashi, T. Tsuru, Steam recovery via nanoporous and subnanoporous organosilica membranes: The effects of pore structure and operating conditions, *Sep. Purif. Technol.* 275 (2021) 119191. <https://doi.org/10.1016/j.seppur.2021.119191>.
- [12] F. Kadir Khan, P.S. Goh, A.F. Ismail, W.N.F. Wan Mustapa, M.H.M. Halim, W.K. Soh, S.Y. Yeo, Recent Advances of Polymeric Membranes in Tackling Plasticization and Aging for Practical Industrial CO₂/CH₄ Applications—A Review, *Membranes (Basel)*. 12 (2022) 71. <https://doi.org/10.3390/membranes12010071>.
- [13] Y. Han, W.S.W. Ho, Polymeric membranes for CO₂ separation and capture, *J. Memb. Sci.* 628 (2021) 119244. <https://doi.org/10.1016/j.memsci.2021.119244>.
- [14] R.A. Roslan, W.J. Lau, A.K. Zulhairun, P.S. Goh, A.F. Ismail, Improving CO₂/CH₄ and O₂/N₂ separation by using surface-modified polysulfone hollow fiber membranes, *J. Polym. Res.* 27 (2020). <https://doi.org/10.1007/s10965-020-02104-6>.
- [15] L.M. Robeson, The upper bound revisited, *J. Memb. Sci.* 320 (2008) 390–400. <https://doi.org/10.1016/j.memsci.2008.04.030>.
- [16] J. Ahn, W.J. Chung, I. Pinnau, M.D. Guiver, Polysulfone/silica nanoparticle mixed-matrix membranes for gas separation, *J. Memb. Sci.* 314 (2008) 123–133. <https://doi.org/10.1016/j.memsci.2008.01.031>.
- [17] A. Ehsani, M. Pakizeh, Synthesis, characterization and gas permeation study of ZIF-11/Pebax® 2533 mixed matrix membranes, *J. Taiwan Inst. Chem. Eng.* 66 (2016) 414–423. <https://doi.org/10.1016/j.jtice.2016.07.005>.
- [18] M. Pakizeh, S. Hokmabadi, Experimental study of the effect of zeolite 4A treated with magnesium hydroxide on the characteristics and gas-permeation properties of polysulfone-based mixed-matrix membranes, *J. Appl. Polym. Sci.* 134 (2017) 1–7. <https://doi.org/10.1002/app.44329>.
- [19] M.B. Mohamad, Y.Y. Fong, A. Shariff, Gas Separation of Carbon Dioxide from Methane Using Polysulfone Membrane Incorporated with Zeolite-T, *Procedia Eng.* 148 (2016) 621–629. <https://doi.org/10.1016/j.proeng.2016.06.526>.
- [20] T.L. Chew, S.H. Ding, P.C. Oh, A.L. Ahmad, C.-D. Ho, Functionalized KIT-6/Polysulfone Mixed Matrix Membranes for Enhanced CO₂/CH₄ Gas Separation,

Polym. . 12 (2020). <https://doi.org/10.3390/polym12102312>.

Appendix B

RE: Point-by-point response for revision to manuscript RSOS-211371.R1

Title: Selectivity Improvement of Polysulfone-Zeolite Templated Carbon Membrane by Annealing and Coating Treatment for CO₂/CH₄ and H₂/CH₄ Separation

Authors: Nurul Widiastuti, Irmariza Shafitri Caralin, Alvin Rahmad Widyanto, Rika

Wijiyanti, Triyanda Gunawan, Zulhairun Abdul Karim, Mikihiro Nomura, Yuki Yoshida

April 29th, 2022

Response to Reviewers' comments:

We would like to thank the Editor and the Reviewers for their constructive comments and suggestions to help us improve our research and the quality of this manuscript. We have carefully considered all the Reviewer's comments and have revised the manuscript to address their concerns. To aid in the reviewing process, we have replied to all the comments on a point-by-point basis and highlighted the revised sections of the main manuscript with blue font color. We hope the manuscript can now be accepted for publishing in Royal Society Open Science.

On behalf of the authors, and with kind regards,

Dr. Nurul Widiastuti (corresponding author for the submission)

Response to Reviewers of Royal Society Open Science

Reviewer #1:	
1	The title might be a bit confusing as the reader might confuse with carbon membranes (in fact this is a PSF/ZTC MMM)
	Response We highly appreciate the reviewer's constructive comment. We have been changed the title becomes "Annealing and TMOS coating on PSF/ZTC mixed matrix membrane for enhanced CO₂/CH₄ and H₂/CH₄ separation"
2	The title indicated the enhancement for CO ₂ /CH ₄ separation by annealing and coating treatment. However, from Table 5, the reviewer cannot clearly see the improvement at least for CO ₂ /CH ₄ separation, for some cases, it decreases and the authors may not fully address my comment why CO ₂ /CH ₄ selectivity is lower than 1. what is the transport mechanism?
	Response We highly recognize the reviewer's positive comments, our response to your comments below: According to gas separation performance data in Table 2, annealed at 190 °C and TMOS coating with 0.03 mol produce CO₂/CH₄ selectivity improvement by 331%, from 1.37 to 5.90. However, different TMOS coating concentrations (0.01 and 0.05 mol) on annealed MMM exhibited low CO₂/CH₄ selectivity, which are 1.00 and 0.93, respectively. The

low selectivity (similar to CO_2/CH_4 Knudsen selectivity of 0.6) indicates that the most gas diffusion contributor is Knudsen diffusion, in which the gas permeance was inversely related to the molecular weight of the penetrated species [1]. In addition, the gas diffusion depended on the mean free path length as the average distance traveled by a gas molecule before colliding with another gas molecule [2]. Thus, it supposes the reason why CO_2/CH_4 selectivity on the membrane with 0.01 mol TMOS coating and annealed at 190 °C below one. Additionally, its selectivity is less than annealed at 190 °C MMM (1.36). It suggests that membrane surfaces are typically covered by TMOS coating, resulting in tighter polymer chain packing and a decrease in free volume.

Interestingly, there is an additional factor that leads to the poor selectivity of MMM with 0.05 mol TMOS coating and annealed at 190 °C. Excess TMOS concentration would diminish selectivity due to the multilayer, which can reduce adhesion between the polymer matrix and filler particles [3]. Furthermore, another diffusion mechanism that gives a major impact is solution diffusion due to the dense structure of the membrane after annealing.

Therefore, we add the discussion below.

Page 10, Line 24-26.

Moreover, its selectivity is less than annealed at 190 °C MMM (1.36). It suggests that membrane surfaces are typically covered by TMOS coating, resulting in tighter polymer chain packing and a decrease in free volume.

Page 11, Line 25-42.

The proposed diffusion mechanism on MMM PSF/ZTC annealed at 190 °C and TMOS coating is a combination of Knudsen diffusion, surface flux, and solution diffusion. Different TMOS coating concentrations (0.01 and 0.05 mol) on annealed MMM exhibited low CO_2/CH_4 selectivity, namely 1.00 and 0.93, respectively. The low selectivity (similar to CO_2/CH_4 Knudsen selectivity of 0.6) indicates that the major gas diffusion contributor is Knudsen diffusion, in which the gas permeance is inversely related to the molecular weight of the penetrated species [1]. Furthermore, gas diffusion is influenced by the mean free path length, which is the average distance travelled by a gas molecule before colliding with another gas molecule [2]. Additionally, a similar trend was observed for H_2/CH_4 separation for 0.01 and 0.05 mol TMOS coating concentration on annealed MMM, which means its selectivity is close to the H_2/CH_4 Knudsen selectivity (2.83). Hence, the Knudsen diffusion mechanism exerts a crucial influence on the gas separation mechanism. On the other hand, membrane selectivity beyond Knudsen selectivity was observed on an annealed MMM coating with 0.03 mol TMOS. Because the optimal silane coating on the membrane surface provides an electrical charge distribution on the membrane, it implies that the electric charge produces a difference in gas separation behavior [4]. Takahashi et al. studied that at the surface of the porous alumina structure, oppositely charged atoms promote a physical interaction between the CO_2 molecule and the silane coupling agent [4]. Due to the fact that the CH_4 molecule is non-polar, while CO_2 has a polarity, thus providing a higher permeation of CO_2 molecules than CH_4 . Therefore, the other diffusion mechanism that makes a contribution is surface flux or facilitated diffusion.

Reviewer #2: The author is familiar with is manuscript content and revises it carefully and completely with a serious attitude. The structure of the article is clear. Recommended for acceptance.

Response

Thank you for your positive feedback. We highly appreciate it.
--

REFERENCES

- [1] Y.-T. Lin, M.-Y. Wey, H.-H. Tseng, Highly Permeable Mixed Matrix Hollow Fiber Membrane as a Latent Route for Hydrogen Purification from Hydrocarbons/Carbon Dioxide, *Membranes (Basel)*. 11 (2021) 865. <https://doi.org/10.3390/membranes11110865>.
- [2] M. Yoshimune, High Selective Carbon Membranes, *Encycl. Membr.* (2015) 1–4. https://doi.org/10.1007/978-3-642-40872-4_2138-1.
- [3] R.A. Roslan, W.J. Lau, A.K. Zulhairun, P.S. Goh, A.F. Ismail, Improving CO₂/CH₄ and O₂/N₂ separation by using surface-modified polysulfone hollow fiber membranes, *J. Polym. Res.* 27 (2020). <https://doi.org/10.1007/s10965-020-02104-6>.
- [4] T. Takahashi, R. Tanimoto, T. Isobe, S. Matsushita, A. Nakajima, Surface modification of porous alumina filters for CO₂ separation using silane coupling agents, *J. Memb. Sci.* 497 (2016) 216–220. <https://doi.org/10.1016/j.memsci.2015.09.007>.